# Molecular motor tug-of-war regulates elongasome cell wall synthesis dynamics in *Bacillus subtilis*

Stuart Middlemiss [1] ✉, Matthieu Blandenet[1], David M. Roberts [2], Andrew McMahon [2], James Grimshaw [1], Joshua M. Edwards [1,2], Zikai Sun[3], Kevin D. Whitley [1], Thierry Blu [3,4], Henrik Strahl [1] ✉ & Séamus Holden [1,2] ✉

Most rod-shaped bacteria elongate by inserting new cell wall material into the inner surface of the cell sidewall. This is performed by class A penicillin binding proteins (PBPs) and a highly conserved protein complex, the elongasome, which moves processively around the cell circumference and inserts long glycan strands that act as barrel-hoop-like reinforcing structures, thereby giving rise to a rod-shaped cell. However, it remains unclear how elongasome synthesis dynamics and termination events are regulated to determine the length of these critical cell-reinforcing structures. To address this, we developed a method to track individual elongasome complexes around the entire circumference of *Bacillus subtilis* cells for minutes-long periods using single-molecule fluorescence microscopy. We found that the *B. subtilis* elongasome is highly processive and that processive synthesis events are frequently terminated by rapid reversal or extended pauses. We found that cellular levels of RodA regulate elongasome processivity, reversal and pausing. Our single-molecule data, together with stochastic simulations, show that elongasome dynamics and processivity are regulated by molecular motor tug-of-war competition between several, likely two, oppositely oriented peptidoglycan synthesis complexes associated with the MreB filament. Altogether these results demonstrate that molecular motor tug-of-war is a key regulator of elongasome dynamics in *B. subtilis*, which likely also regulates the cell shape via modulation of elongasome processivity.

Almost all bacteria are encased by a peptidoglycan-based cell wall, which is essential for their survival. To maintain a robust cell wall during growth and division, bacterial cell wall synthesis proteins must accurately and reliably expand and remodel a precisely shaped structure more than 100 times their size. Due to the high internal turgor of a bacterial cell, major errors in cell wall synthesis trigger lethal cell lysis.

For this reason, the cell wall synthesis machinery is the principal target of many first-line antibiotics such as β-lactams, as well as last resort antibiotics such as vancomycin and daptomycin that are used to treat infections caused by multidrug-resistant pathogens. A better understanding of the biophysical principles of cell wall synthesis is therefore critical for deciphering how this highly successful class of antibiotics

[1]Centre for Bacterial Cell Biology, Biosciences Institute, Faculty of Medical Sciences, Newcastle University, Newcastle upon Tyne, UK. [2]School of Life Sciences, University of Warwick, Gibbet Hill Campus, Coventry, UK. [3]Department of Electronic Engineering, The Chinese University of Hong Kong, Hong Kong, China. [4]Dept of Electrical Engineering, National Taiwan University, Taipei City, Taiwan. ✉ e-mail: stuart.middlemiss@newcastle.ac.uk; h.strahl@newcastle.ac.uk; seamus.holden@warwick.ac.uk

can induce bacterial cell death, for developing the next generation of cell-wall-targeting antibiotics, as well as developing countermeasures against the adaptive processes bacteria utilise to evade those already in clinical use.

Most rod-shaped bacteria, including key model organisms such as Gram-positive *B. subtilis* and Gram-negative *Escherichia. coli*, elongate by inserting new cell wall material into the inner surface of the cell wall. This is performed by class A penicillin binding proteins (PBPs) and a highly conserved protein complex, the elongasome, which inserts long peptidoglycan strands circumferentially around the cell, giving rise to a rod-shaped cell morphology[1,2].

Gram-positive rod-shaped bacteria such as *B. subtilis* have a single cytoplasmic membrane surrounded by a thick multi-layered peptidoglycan cell wall. Elongasome-driven cell wall synthesis is performed by a coordinated action of two enzymes: the glycosyltransferase RodA, which polymerizes glycan strands, and a cognate class B transpeptidase (PBP2A or PBPH in *B. subtilis*), which attaches new strands to the existing cell wall[3]. These proteins, together with additional regulatory factors, are associated with the actin-homologue MreB, which forms antiparallel double-filament structures[4]. Each double filament is around 170-nm long[5] and monomers are around 5-nm long[4], suggesting that the average MreB double filament consists of around 68 subunits. These cytoskeletal structures guide peptidoglycan insertion perpendicular to the long axis of the cell[6]. *B. subtilis* also encodes two functionally redundant MreB homologues, Mbl and MreBH, which copolymerise with MreB[7]. As continuous glycan chains can stretch less than cross-linked peptides, circumferentially oriented glycan strands reinforce the cell sidewall and thereby establish a rod-like cell shape[1,2]. The overall level of elongasome-driven cell wall synthesis plays a major role in establishing both the all overall rod shape morphology and the specific cell diameter. Hereby high levels of elongasome-driven cell wall synthesis lead to stiff, narrow, rod-shaped cells whereas low levels lead to soft, wide, spherical cells[1,2].

The processive motion of the elongasome is driven by peptidoglycan synthesis[8–10]. It is likely that the initial length of elongasome-synthesised glycan strands is determined by the processivity of the elongasome, i.e., the distance around the cell circumference that elongasome moves in one direction without interruption. This should reflect an individual processive synthesis event. As the primary function of the elongasome is to maintain rod-shape and elongate the cell sidewall by inserting circumferential glycan strands, we hypothesised that elongasome processivity, and thus the length of elongasome-synthesised glycan strands, is likely to have substantial effect on cell wall stiffness and thereby cell shape. Put simply – if elongasome synthesised glycans act as reinforcing structures similar to metal hoops around a wooden barrel, the length of those reinforcing structures should determine the stiffness of the cell's short axis.

To test these hypotheses, we developed a new method to track individual elongasome complexes around the entire cell circumference for minutes-long periods. We found that the *B. subtilis* elongasome is highly processive and exhibits frequent reverses and pauses. Intriguingly, we found that cellular levels of RodA regulate elongasome processivity, reversal and pausing. Together with stochastic simulations, our single-molecule data support an end-binding tug-of-war model where competition between two opposing peptidoglycan synthesis complexes, bound to each end of the symmetrical MreB double filament, determine elongasome dynamics and processivity. Elongasome tug-of-war may also regulate *B. subtilis* cell shape via modulation of elongasome processivity. Our data demonstrate that molecular motor tug-of-war is a key regulator of elongasome dynamics, which may also play a major role in bacterial cell shape control in *B. subtilis*.

## Results

### The *B. subtilis* elongasome is highly processive and frequently reverses and pauses

Previously, it was only possible to track elongasomes that are performing cell wall synthesis over distances less than about 500 nm, owing to the geometrical limitations of total internal reflection fluorescence (TIRF) imaging, which is usually used for such measurements but which only illuminates a small fraction of the cell circumference. Previous estimates of elongasome processivity have been in the range of 400–600 nm[11], suggesting that these measurements were limited by the shallow illumination depth of the technique. We define processivity as the distance travelled in a continuous motion by an elongasome around the circumference of the cell until pausing, reversal or another terminating event. In a kymograph, this corresponds to the displacement of signal around the cell circumference over time (Supplementary Fig. 1).

To address this limitation, we combined single-molecule tracking with VerCINI (vertical cell imaging by nanostructured immobilisation), a method we developed to orient rod-shaped cells perpendicular to the microscope imaging plane[12]. We used single-molecule VerCINI (smVerCINI) to focus on a slice of the bacterial cell sidewall approximately 0.5-µm thick (determined by the microscope objective depth of field), and tracked individual membrane bound MreB molecules in live *B. subtilis* cells using a previously characterised functional, native-locus MreB-HaloTag fusion[6] (Supplementary Fig. 2). We used a sub-stoichiometric labelling concentration of the bright cell-permeable HaloTag ligand JF549 (a JaneliaFluor dye[13]) to sparsely label individual MreB molecules within membrane-bound MreB filaments (Fig. 1a). Because MreB motion is circumferential[5,6], MreB filament dynamics are mostly constrained to within the VerCINI focal plane, allowing long term imaging of MreB filament dynamics (Fig. 1a).

We tracked MreB molecules using long (500 ms) camera exposure times such that freely diffusive molecules were not detected, allowing us to exclusively analyse MreB molecules assembled within membrane bound MreB filaments, while simultaneously reducing effective photobleaching rate using long (6 s) strobe intervals. We analysed the data using kymographs (see schematic Supplementary Fig. 1, also definitions Supplementary Table 1). We found that MreB filaments remain assembled at the membrane for extended periods of time, frequently reverse direction and sometimes pause for extended periods (Fig. 1b, c, Supplementary Movies 1–3). As MreB filament motion is dependent on peptidoglycan synthesis, motile MreB filaments should correspond to fully assembled elongasome complexes actively engaged in peptidoglycan synthesis. Paused MreB filaments could correspond either to filaments in which some or all the other critical elongasome components are missing/ unbound, or to fully assembled complexes currently not engaged in peptidoglycan synthesis.

We determined the binding lifetime of MreB subunits within filaments and the JF549 photobleaching lifetime using the stroboscopic illumination method of Gebhardt and coworkers[14]. We measured the apparent binding lifetime of MreB molecules by single-molecule tracking and systematically increased the total time interval between frames (strobe interval), while keeping the exposure time and thus total light dose constant (Fig. 1d, e). As the strobe interval increased, the fraction of time for which the molecule was illuminated decreased thus increasing the effective photobleaching lifetime. This not only allows molecules to be tracked for longer but also allows accurate determination of both molecule binding lifetime time and dye photobleaching lifetime by fitting an equation describing the relationship between those quantities, the strobe interval, and the apparent binding lifetime (Fig. 1e, Methods).

We measured the MreB binding lifetime as 128 s [95% CI: 109, 164] showing that the MreB filaments remain assembled at the membrane for extended periods of time. This measurement represents a lower bound on the lifetime of both assembled MreB filaments, as it likely

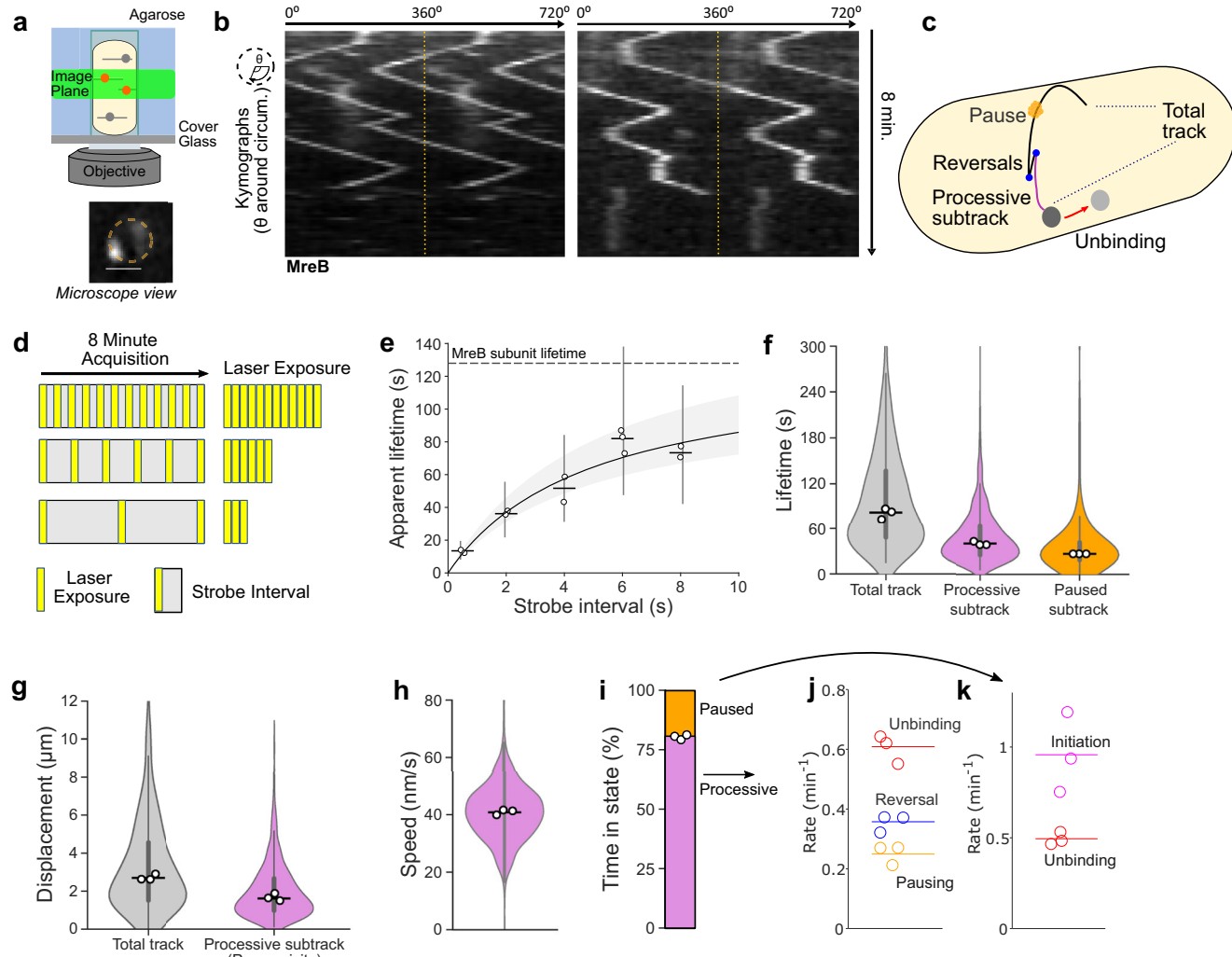

**Fig. 1 | Single-molecule VerCINI (smVerCINI) measurements of MreB dynamics.**
**a** Principle of smVerCINI. Grey lines represent individual MreB filaments, orange circles indicate MreB subunits sparsely labelled with JF549 fluorophore. **b** Exemplar kymographs of MreB filament dynamics. Kymographs are measured around the cell circumference. Two full revolutions around the cell (0–720°) are plotted side-by-side to resolve filament trajectories that pass 0°/360°, separated by yellow dotted lines. **c** Cartoon illustrating different types of MreB filament dynamics observed. **d** Principle of stroboscopic illumination: increasing intervals between constant illumination time reduce photobleaching and allow estimation of photobleaching rate and molecule unbinding rate. **e** Stroboscopic illumination plot of strobe interval versus apparent MreB subunit lifetime. Black line: non-linear fit of Gebhardt model (Methods). Grey shaded area: 95% CI on fitted model. Horizontal dashed line: estimated MreB subunit lifetime. Vertical lines: IQR of apparent lifetimes. Fitted

subunit lifetime $\tau_{off} = 128$ s [95% CI: 109, 164], photobleaching lifetime, $\tau_{bl} = 13$ s [95% CI: 11, 16]. **f, g** MreB filament lifetime and circumferential track displacement for individual tracks and processive or static subtracks. Total tracks refer to the whole observed elongasome trajectory, whereas subtracks refer to regions within a track whose motion type remains constant, i.e. constant processive motion or pausing. **h** MreB filament speed for processive subtracks. **i** Time MreB filaments spend in each motion state. Single-molecule switching rates for MreB processive subtracks (**j**) and static subtracks (**k**). Measurements in **b, f–k** performed at 6 s strobe interval. White filled circles: median of biological replicates. Horizontal lines: median of all data points. Violin plots: thick error bar lines indicate IQR, thin error bar lines indicate 1.5× IQR. Strain used: *B. subtilis* SM01 (*mreB-HaloTag, Δhag*). Sample sizes and numbers of experimental replicates are listed in Supplementary Table 7.

represents a combination of MreB filament unbinding from the membrane, dissociation of MreB subunits from the MreB filament and occasional migration of the elongasome complexes beyond the microscope depth of field. We also measured the JF549 bleaching lifetime as 13 s [95% CI: 11, 16].

We next characterised MreB motility by smVerCINI. We chose a strobe interval of 6 s, which extended the effective photobleaching lifetime 12-fold to 156 s [95% CI: 132, 192] (Fig. 1e), longer than the median observed MreB subunit lifetime of 128 s [95% CI: 109, 164]. This allowed direct measurements of MreB single-molecule switching kinetics. We found that MreB filaments are motile 81 % of the time, [Range: 79–81, *n* = 3], and immobile (paused) the rest of the time (Fig. 1i). MreB filaments frequently switch between motile and paused states, and motile MreB filaments frequently switch direction

(reversal), which likely corresponds to initiation of PG synthesis in the opposite direction (Fig. 1i–k), Supplementary Fig. 10d). The median lifetimes of both the processive and paused motility states were substantial: 40.5 s [95% CI: 39.0, 43.0] and 27.0 s [95% CI: 24.0, 29.5], respectively (Fig. 1f).

While elongasome pauses and reversals have been observed before, they were previously thought to be rare events[11,14], likely due to elongasome trajectory truncation due to TIRF imaging. Strikingly however, we found that 51 % [Range: 48–52, *n* = 3] of elongasome processive synthesis events are terminated by changes in elongasome dynamics - reversal or pausing – rather than by elongasome disassembly or MreB dissociation (Fig. 1j). These data thus demonstrate that bidirectional elongasome motility is a central feature of elongasome dynamics. Furthermore, given that so many synthesis events

terminate due to changes in motility state rather than disassembly/ dissociation, elongasome bidirectional motility must play a significant role in determining elongasome processivity. The positions of elongasome reversals and pauses did not show any obvious pattern, suggesting that such switching dynamics are caused by intrinsic factors within the elongasome, rather than local heterogeneity in the cell wall template.

Using smVerCINI, we found that MreB filaments, and therefore *B. subtilis* elongasomes, are highly processive. Complete MreB tracks were found to contain multiple substates, where MreB was observed either to move processively in the same direction at constant speed – corresponding to active cell wall synthesis[8–10], or to pause for extended periods. Processive subtracks were found to have a median displacement of 1.61 μm [95% CI: 1.51, 1.69] (Fig. 1g), or approximately 180° around the cell circumference (Supplementary Fig. 3). This is substantially greater than the 0.5 μm previously estimated by TIRF[11]. We found that MreB moved at constant speed of 41 nm/s [95% CI: 40, 41], independent of the processive subtrack lifetime (Fig. 1h, Supplementary Fig. 4b). The speeds we observed were similar to those measured previously[8,9]. As this processivity likely determines the initial length of elongasome-synthesised glycan strands, these data support a model where the elongasome-synthesised peptidoglycan strands act as major reinforcing structural elements in the cell sidewall, much like hoops around a barrel.

Given that cell wall synthesis rates correlate with cell growth, we wondered how cell growth rate affected elongasome processivity. Surprisingly, we found that large 3-fold changes in growth rate have only a modest effect on elongasome processivity and switching dynamics (Supplementary Fig. 5). Deletion of *mltG*, which has been proposed as a possible terminator of peptidoglycan synthesis[15] also showed minimal effect on elongasome processivity (Supplementary Fig. 6).

## RodA expression level regulates bidirectional motility and elongasome processivity

It was previously proposed that elongasome reversals could be caused by molecular motor tug-of-war, whereby two or more RodA-PBP2A/ PBPH synthesis complexes attached to a symmetrical MreB-filament pull in opposite directions, similar to eukaryotic organelle transport[16]. However, this model was not widely accepted as bidirectional motility of elongasomes was until now assumed to be a rare, inconsequential feature of elongasome dynamics, and also due to other limitations in the original tug-of-war model, outlined in the next section.

Given our frequent observation of elongasome reversals and pauses, which are strongly reminiscent of eukaryotic molecular motor tug-of-war, we decided to revisit the elongasome tug-of-war model. In the eukaryotic model, cargo such as lipid droplets is transported bidirectionally along microtubules by competing molecular motors which constantly attempt to drag the cargo in opposite directions[17]. In the elongasome tug-of-war model, the molecular motors (peptidoglycan synthases) move processively along the cell wall track, which is analogous to the microtubule track in the eukaryotic model. The cargo in the case of the elongasome is the MreB filament. PG synthesis results in processive motion of both the elongasome synthases and the MreB filament relative to the cell wall (Supplementary Fig. 7). Motor competition in the case of the elongasome corresponds to attempts by synthesis complexes to initiate cell wall synthesis in the opposite direction to the current direction of elongasome cell wall synthesis (Supplementary Fig. 7).

We set out to test whether elongasome complexes might indeed participate in molecular motor tug-of-war, and to determine whether tug-of-war mediated reversals might thereby determine elongasome processivity. We titrated cellular levels of the elongasome transglycosylase RodA in a strain with inducible expression from the native locus (*rodA':P_{spac}-rodA*[18]), and measured elongasome dynamics by

smVerCINI of MreB-HaloTag. At low *rodA* expression levels, we observed that elongasomes exhibited extended pauses, infrequent reversals, and high speed (Fig. 2, Supplementary Movies 4–8). In contrast, at high *rodA* expression levels, pauses were rare, reversals more frequent and speeds lower (Fig. 2, Supplementary Movies 9–13). MreB pausing rate decreased 0.43-fold (−0.13 min⁻¹ difference [95% CI: −0.17,−0.10]), reversal rate increased 1.1-fold (0.20 min⁻¹ difference [95% CI: 0.17,0.24]) and motile MreB speed decreased 0.39-fold (−20 nm s⁻¹ difference [95% CI: −20,−19]) at high vs low *rodA* expression levels (1 mM IPTG vs 100 nM IPTG induction, Fig. 2d, f). No change was detected in MreB unbinding rate (0.01 min⁻¹ difference [95% CI: −0.03, 0.06] (Fig. 2d), consistent with MreB movement being driven by peptidoglycan synthesis rather than MreB polymerisation/depolymerisation. Intriguingly, MreB processivity decreased 0.44-fold, (−0.91 μm difference [95% CI: −1.0, −0.78]) between high and low *rodA* expression, and the processivity of cells expressing *rodA* from the native promotor was near the mid-point of this range (Fig. 2g).

Together, these data show that elongasome dynamics and processivity are sensitively regulated by the cellular concentration of RodA. The data are consistent with a model where increased RodA levels lead to more active synthesis complexes bound to each MreB filament, thereby causing more frequent incidences of tug-of-war between oppositely oriented synthesis complexes, leading to frequent and rapid elongasome reversals. These data are also consistent with a model where high levels of tug-of-war reduce the overall elongasome processivity by more frequent reversals and reduced average elongasome speed due to drag from competing synthesis complexes.

One possible alternative model is that elongasome reversals could be caused by collisions between two elongasome complexes. TIRF-structured illumination microscopy (SIM) has previously been used to observe isolated MreB filaments undergoing reversals without any other filaments nearby to collide with, inconsistent with the collision model[8,19]. In a third model, elongasome reversals could be caused by interactions with existing peptidoglycan; if, for example, glycans oriented at certain angles could act as effective barriers to the elongasome. While it is possible that interactions with the peptidoglycan may play some role in elongasome bidirectional motility, that model does not explain why elongasome reversal rate increases, or why pausing rate decreases, as *rodA* expression level is increased.

We also tested how single knockouts of the redundant elongasome transpeptidases PBP2A (*pbpA*) and PBPH (*pbpH*) affected elongasome dynamics. Deletions of these genes had little effect on elongasome switching kinetics, speed or processivity (Supplementary Fig. 6, Supplementary Table 2). This is consistent with previous observations whereby overexpression of individual elongasome transpeptidases alone did not affect cell width[2]. In addition, the elongasome transpeptidase in *E. coli* is produced to levels where most of the protein is freely diffusive and not bound to the elongasome[20]. Together, these results support a model where the elongasome transpeptidases are in excess compared to RodA, and thus not the limiting factor in elongasome activity. Our data show that RodA concentration, or its assembly to an elongasome complex, is a principal factor controlling the concentration of active elongasome synthesis complexes within the cell any given time.

We also considered whether cellular levels of lipid II could play a role in regulating elongasome dynamics. Overexpressing *murAA*, an enzyme at the beginning of the lipid II synthesis pathway has previously been observed to lead to increased lipid II cellular levels and altered cell growth rate in minimal medium[21]. However, upon overexpression of *murAA* in minimal medium we did not observe changes in cell growth rate and morphology, elongasome processivity or elongasome speed relative to native *murAA* expression levels (Supplementary Fig. 8, Supplementary Table 2), suggesting that lipid II levels do not rate limit cell growth under our experimental conditions. Additionally, if lipid II levels dropped substantially upon *rodA*

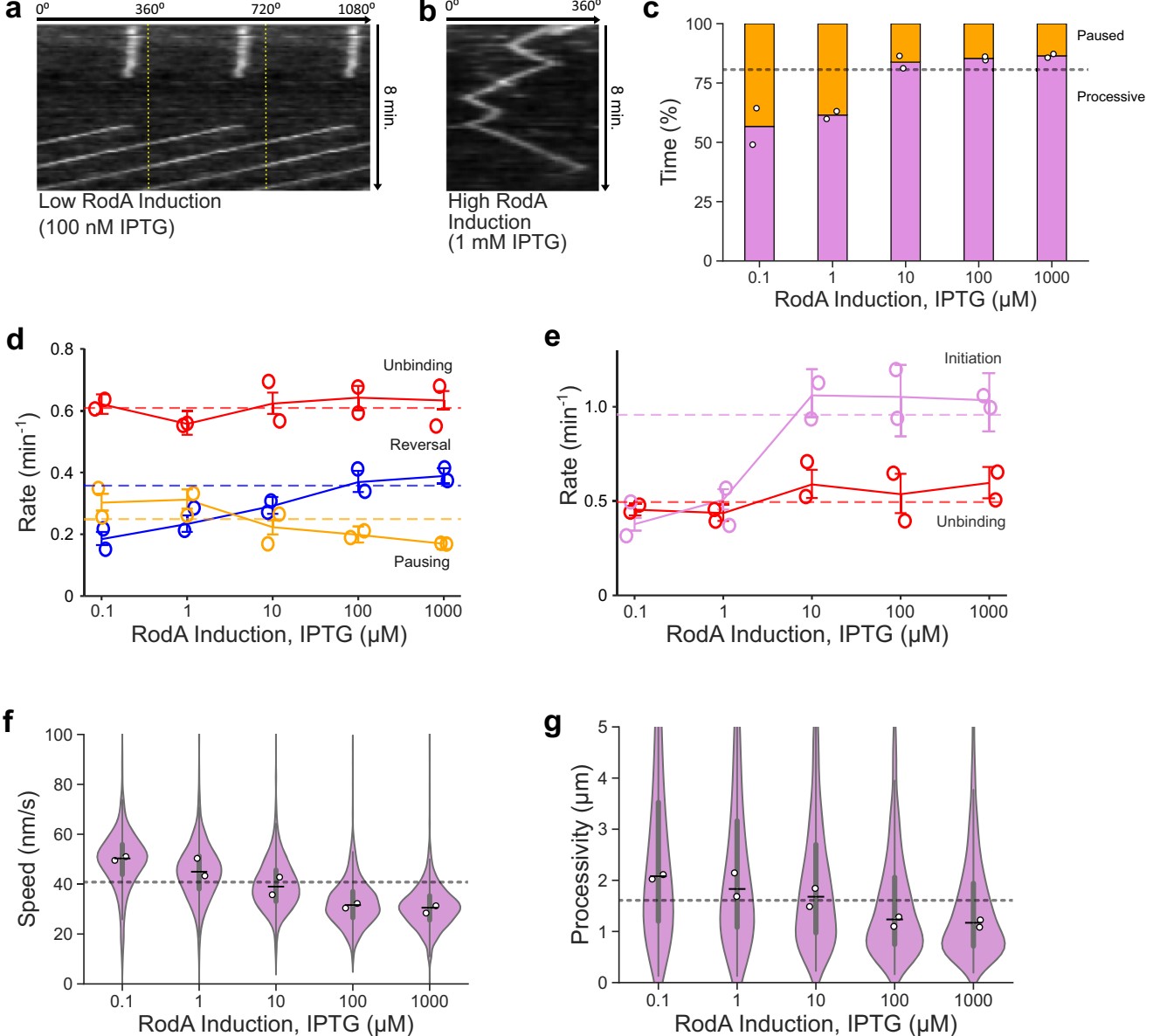

**Fig. 2 | Effect of cellular levels of RodA on MreB dynamics.** Exemplar kymographs of MreB filament dynamics at low (**a**) and high (**b**) RodA levels achieved through expression from an IPTG-inducible promoter. Kymographs are measured around the cell circumference. More examples are shown in Supplementary Fig. 11. **c** Time MreB filaments spend in each motion state as a function of rodA expression level. Single-molecule switching rates for MreB processive subtracks (**d**) and static subtracks (**e**). Solid coloured lines represent medians of all data points for each condition. Error bars represent 95% CI. **f**, **g** MreB filament speed and processivity for processive subtracks. White filled circles: median of biological replicates. Horizontal dashed lines: value of each parameter (eg rate, speed) at native rodA expression level in strain SM01 (*mreB-HaloTag, Δhag*). Narrow horizontal lines: median of all data points. Violin plots: thick error bar lines indicate IQR, thin error bar lines indicate 1.5× IQR. Strain used: *B. subtilis* SM28 (*mreB-HaloTag, Pspac-rodA, Δhag*). Further quantification in Supplementary Fig. 12. Sample sizes and numbers of experimental replicates are listed in Supplementary Table 7. Effect sizes are listed in Supplementary Table 2.

overexpression because of the presence of more enzymes to consume the substrate, the likely consequences for elongasome dynamics would be that elongasome complexes (i) pause or terminate synthesis more frequently and/ or (ii) unbind or disassemble more frequently. We observed neither of these effects in our data (Fig. 2d). Lastly, substrate limitation cannot plausibly explain the observed increased rate of elongasome reversals upon *rodA* overexpression, as this requires the same or higher levels of elongasome activity, not less. These data indicate that cellular lipid II levels are unlikely to play a major role in regulating elongasome bidirectional motility.

**Stochastic simulations show that an end-binding tug-of-war model can explain experimentally observed effects of RodA**

**expression level on elongasome processivity and bidirectional motility**

Previously, it was proposed that elongasome tug-of-war between many synthases bound along the entire MreB filament might regulate elongasome dynamics[19]; a scenario which we term the *unlimited binding* elongasome tug-of-war model (Fig. 3b). However, this model was not widely accepted as the bidirectional motility of elongasomes was until now assumed to be a relatively unimportant feature of elongasome dynamics, mostly due to underestimation of elongasome pausing and reversal rates[11,14]. The *unlimited-binding* tug-of-war model also assumed a large numbers of competing synthesis complexes bound to a single MreB filament, which has since been found to be unlikely[5] and predicted a strong dependence of elongasome speed upon MreB

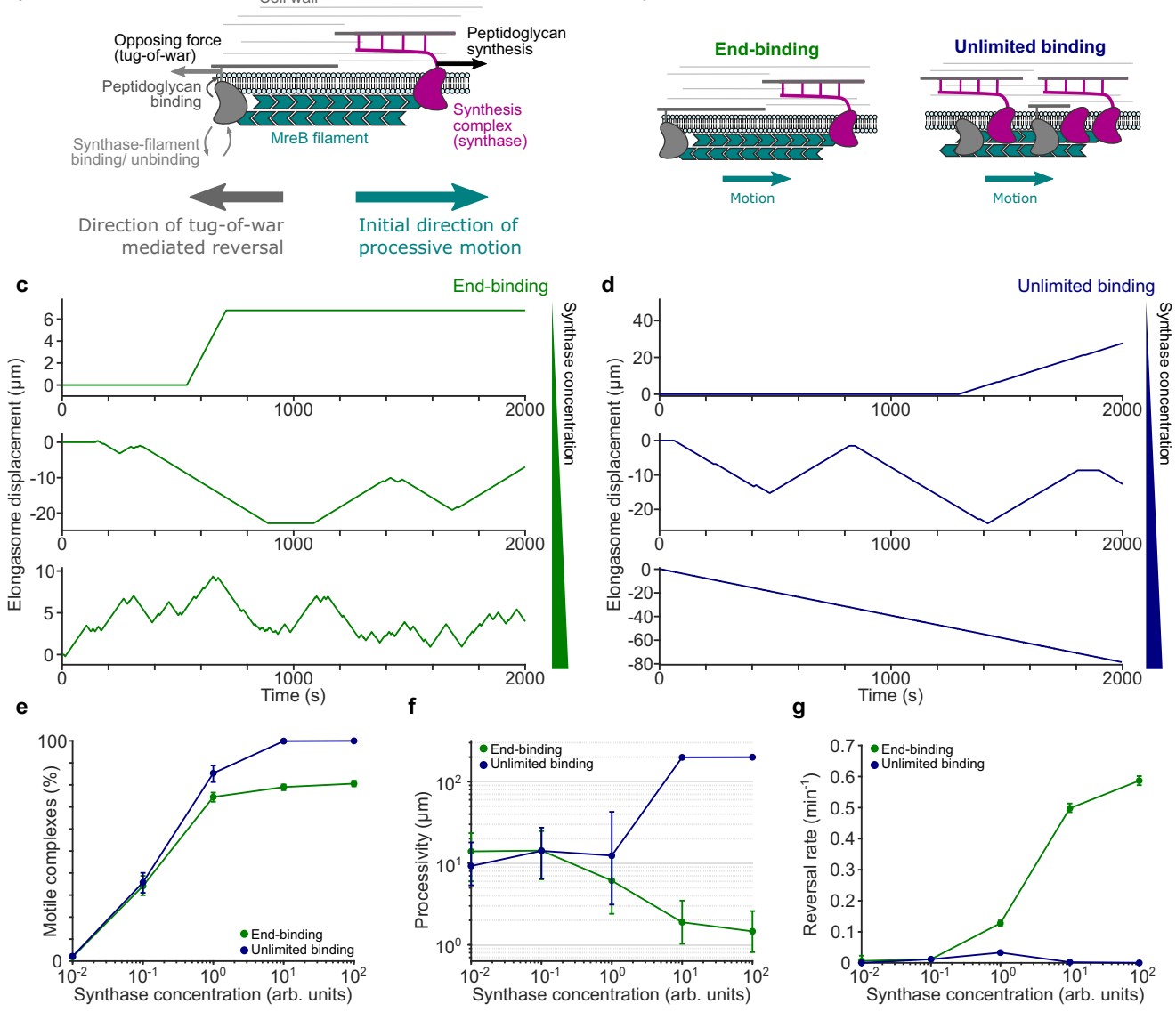

**Fig. 3 | Simulations of elongasome tug-of-war dynamics. a** Cartoon of elongasome complex dynamics as implemented in the stochastic model. **b** Illustration of the two models of elongasome tug-of-war tested: end-binding, where only one synthesis complex can bind to each end of the antiparallel MreB double filament, in opposite directions; unlimited binding, where multiple synthesis complexes can bind along the MreB filament. **c, d** Examples of simulated elongasome dynamics at low ($10^{-2}$ arb. Units), intermediate ($10^{0}$ arb. Units) and high ($10^{2}$ arb. Units) synthesis complex (synthase) concentrations for end-binding and unlimited binding models. **e–g** Fraction of motile elongasomes, their processivity and reversal rate as a function of synthesis complex concentration for both models. Solid coloured lines with filled circles, sample medians; vertical lines, 95% CI. Number of simulation replicates for each condition, 100. Further quantification in Supplementary Fig. 13.

filament length which was later found not to be the case[5]. Furthermore, previous theoretical work on eukaryotic molecular motors showed that as more molecular motors are added to an *unlimited-binding* tug-of-war scenario, a runaway scenario occurs where reversals become exponentially less likely as a single opposing motor must win the tug-of-war against an ever-greater number of engaged motors[17]. Taken together, the *unlimited-binding* model is not easy to reconcile with our observations that MreB reversal rate increases and processivity decreases at high RodA levels. To address this, and inspired by the observation that MreB forms a symmetric antiparallel double filament[4], we proposed an *end-binding* elongasome tug-of-war model, where at most two synthesis complexes can bind to an MreB filament, one at each end, pointing in opposite directions (Fig. 3b). The *end-binding* model inherently avoids large numbers of synthesis complexes per filament, as well as the highly processive multi-motor scenario. Alternatively, we speculated that the *unlimited-binding* model

might be able to explain our experimental observations if cellular concentrations of elongasome synthesis complex components are low enough to limit the number of synthesis complexes per MreB filament to around 1-2 on average, and thereby mostly avoid the runaway elongasome scenario.

To test these two models, we used Monte Carlo simulations to evaluate whether either the *end-binding* or *unlimited-binding* synthase tug-of-war models are physically plausible mechanisms to regulate elongasome reversal rate and processivity. The simulations are an extension of the Müller, Klumpp, and Lipowsky (MKL) model of eukaryotic cargo transport[17], and assume that multiple RodA-bPBP synthesis complexes can bind to both the MreB filament and the existing cell wall to initiate peptidoglycan synthesis (Fig. 3a, Supplementary Note 1). Synthesis complexes attempting to perform peptidoglycan synthesis in opposite directions will stall and briefly engage in tug-of-war, resulting in either resumption of peptidoglycan

synthesis in the original direction, or reversal and initiation of peptidoglycan synthesis in the opposite direction.

We performed simulations of each model over a range of synthesis complex concentrations, using an extension of the MKL model to allow concentration dependent binding/ unbinding of synthesis complexes from the MreB filament (Fig. 3, Supplementary Fig. 7, Supplementary Note 1). Both models showed extended elongasome pausing and infrequent reversals at low synthesis complex concentrations (Fig. 3), similar to our experimental results (Fig. 2). At intermediate synthesis complex concentration, elongasome reversal rate increased for both the *end binding* and *unlimited binding* models. At high synthesis complex concentrations, the *end binding* model still showed frequent reversals, consistent with our experimental data. However, the reversal rate for the *unlimited binding model* declined rapidly once the average number of bound synthesis complexes increased beyond two (Fig. 3g).

We also found that as synthesis complex concentration increased, the *end binding* model processivity decreased in a manner consistent with our experimental observations. While the *unlimited binding* model shows a transient increase in reversal rate at intermediate synthase concentrations, which could potentially partially reproduce experimental results, even in this regime we were not able to reproduce the experimentally observed increase in processivity as a function of synthase concentration in simulations of the *unlimited binding* model as implemented here. Consistent with previous analyses of the MKL model for eukaryotic motor tug-of-war[17], at high synthase concentration the *unlimited binding* model predicts almost indefinitely processive elongasomes - limited here only by the length of the simulations. This is because individual elongasomes are very unlikely to win a tug-of-war against large numbers of oppositely bound active synthases. This runaway-motor scenario is inconsistent with experimental results.

These data show that an *end-binding* synthase tug-of-war model is a physically plausible model that is sufficient to recapitulate experimentally observed trends in elongasome reversal rate and processivity. Interestingly, this model also makes strong predictions about the structure, location and number of RodA-bPBP complexes on MreB filaments, which could be tested in future to better understand the molecular mechanisms underlying elongasome tug-of-war.

### Cell widening upon *rodA* overexpression may be driven by tug-of-war mediated reduction in elongasome processivity

RodA protein levels have previously been shown to control *B. subtilis* cell width in a non-trivial manner[2]: low or high RodA levels lead to abnormally wide cells, whereas intermediate levels ensure narrower, wild-type-like cell morphology. We confirmed that the cell widening phenotype upon *rodA* overexpression also occurred in the minimal media and culture conditions used in this study (Fig. 4a, Supplementary Fig. 9).

Recent studies support a model where that cellular levels of motile elongasomes determine cell width by controlling the density of newly synthesised circumferentially oriented glycan strands and thus regulating lateral cell wall stiffness. This model predicts that cell width decreases as elongasome synthase concentration increases[2]. However, this model is insufficient to explain the increase in cell width upon overexpression of *rodA*[2]. We found that cell widening upon *rodA* overexpression (1 mM IPTG) is not associated with any detectable change in surface density of motile MreB filaments compared to induction with 10 μM IPTG (0.36 track density μm$^{-2}$ difference [95% CI: −0.54, 1.3]) (Fig. 4b, Supplementary Table 2), which have previously been shown to regulate cell width[2], nor any obvious change in cell growth rate (Supplementary Fig. 9).

Our findings that RodA levels determine elongasome processivity via tug-of-war mediated regulation (Figs. 2 and 3) provide a simple mechanistic model for the complex dependence of cell width on cellular RodA levels. We hypothesise that cells must maintain a balance between: (i) elongasome pausing at low synthase levels, which reduces cellular levels of motile elongasomes; and (ii) tug-of-war at high synthase levels, which reduces elongasome processivity (Fig. 2). Since elongasome synthesised glycan strands act to reinforce the cell sidewall, both the length and total number of elongasome synthesised glycans should determine cell sidewall stiffness and width. Therefore, an optimally stiff, narrow cell wall would be synthesised at intermediate synthase concentration levels, which balances the opposing constraints of density of active elongasomes and elongasome processivity (Fig. 5c, d).

It was previously speculated that cell widening upon *rodA* overexpression could be caused by high levels of disorganised synthesis by RodA-bPBP complexes not bound to the elongasome[2]. However, experimental evidence has not yet been presented for that hypothesis. Further experiments will be required to conclusively determine whether elongasome tug-of-war, off-target RodA-bPBP synthesis or a combination of both drive cell widening upon *rodA* overexpression.

## Discussion

The elongasome plays a central role in cell wall growth and maintenance of cell shape in a wide range of bacteria. In this study, we found

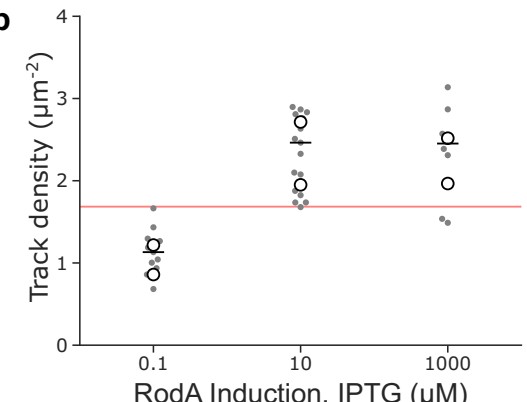

**Fig. 4 | Effect of cellular levels of RodA on cell shape and MreB filament density.** Effect of titration of RodA cellular levels expressed as sole cellular copy from an inducible promotor on (**a**) cell diameter, and (**b**) surface density of motile MreB filaments measured by SIM-TIRF microscopy. Horizontal lines show overall median. Violin plots: thick error bar lines indicate IQR, thin error bar lines indicate 1.5× IQR. White filled circles indicate biological replicates. Grey filled circles indicate medians of each field of view in SIM-TIRF experiments. Red line indicates values for native RodA levels. Strain used: *B. subtilis* SM28 (*mreB-HaloTag, Pspac-rodA, Δhag*). Sample sizes and numbers of experimental replicates are listed in Supplementary Table 7. Effect sizes are listed in Supplementary Table 2.

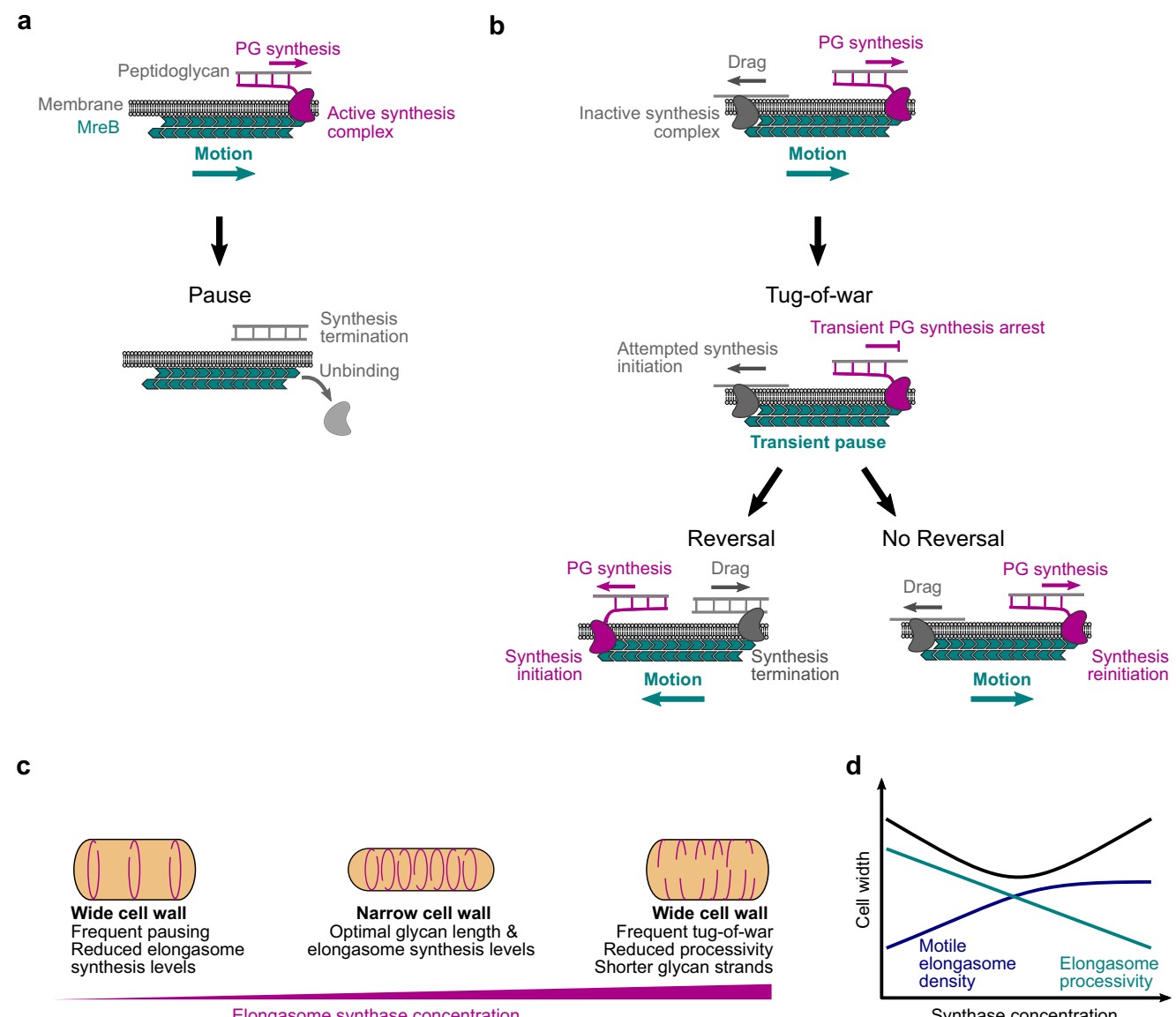

**Fig. 5 | Tug-of-war model of elongasome dynamics and cell size regulation.**
Models for effect of number of bound active synthesis complexes on elongasome dynamics and cell size. Active synthases complexes refer to elongasome core complexes, i.e., active RodA-bPBP pairs plus accessory proteins MreC and MreD. **a** Model for MreB dynamics when a single synthesis complex is bound to the MreB filament. Peptidoglycan synthesis by an active single synthesis complex (purple) drives processive MreB filament motion (teal) until the synthesis complex terminates PG synthesis and unbinds from the MreB filament, leading to pausing of the MreB filament. **b** Model for MreB tug-of-war dynamics, assuming that maximally two synthesis complexes can bind to the MreB filament in opposite orientations. Initially, PG synthesis by the active synthesis complex (purple) drives processive motion of the MreB filament (teal). The oppositely bound inactive synthesis complex (grey) causes drag, possibly through transient interactions with the PG. Upon successful binding of the inactive synthesis complex with the PG, the MreB filament enters a paused tug-of-war state where PG synthesis by the active synthesis complex is transiently arrested. There are two possible outcomes of tug-of-war. If the active synthesis complex wins (no reversal), PG synthesis and MreB

motion is resumed in the original direction. If the inactive synthesis complex wins (reversal), that complex initiates PG synthesis and the formerly active complex terminates synthesis. This leads to reversal of MreB filament motion, with the old lagging edge of the MreB filament becoming the new leading edge. **c, d** Speculative model for effect of elongasome synthase concentration on cell shape. At low concentrations of active elongasome synthases, eg due to the lack of one or more essential components such as RodA in this study, there is both infrequent elongasome PG synthesis and infrequent tug-of-war. This leads to a low density of long elongasome-synthesised PG strands are inserted into the cell wall, resulting in a weaker, wider cell wall. At high concentrations of active elongasome synthases, there is both frequent elongasome PG synthesis and frequent tug-of-war. This leads to a high density of short elongasome-synthesised PG strands inserted into the cell wall, again resulting in a weaker, wider cell wall. At intermediate concentrations of active elongasome synthases, there is frequent elongasome PG synthesis but infrequent tug-of-war. This leads to a high density of long elongasome-synthesised PG strands inserted into the cell wall, resulting in a narrow, optimally strong cell wall.

that *B. subtilis* elongasomes are highly processive with each event covering on average half the cell circumference, thus supporting a model in which elongasome synthesised glycan strands function as major structural elements that reinforce the cell sidewall. We found that bidirectional motility – reversal and pausing – is not a rare curiosity as previously thought but rather a central feature of elongasome

dynamics. We showed that elongasome processivity and bidirectional motility are regulated by molecular motor tug-of-war between multiple elongasome synthesis complexes (Fig. 5a, b). Our data also provide initial evidence for an end-binding tug-of-war between maximally two synthesis complexes bound to opposite ends of the antiparallel MreB filament (Fig. 5a, b). We also found evidence that elongasome

tug-of-war may regulate cell size and shape via modulation of elongasome processivity, and thereby the length of new glycan strands (Fig. 5c, d). These results establish molecular motor tug-of-war acts as a major regulator of bacterial cell wall synthesis activity.

Our study shows that molecular motor tug-of-war regulates MreB-cytoskeleton-associated cell wall synthesis in the model bacterium *B. subtilis*. This phenomenon, previously thought to be exclusive to eukaryotes, is the first of its kind described on a molecular level in bacteria. Similar to bidirectional molecular motor transport in eukaryotes[16], molecular motor tug-of-war enables straightforward tuning of synthase dynamics, and could thereby facilitate rapid regulation of cell wall material properties, by regulating the concentration or activity of the Rod-complex. Tug-of-war mediated bidirectional motility may also allow obstacles in the cell wall to be avoided and peptidoglycan synthesis to be distributed evenly around the surface of the cell wall. Further studies will be required to determine the molecular principles underlying elongasome tug-of-war and the role of molecular motor tug-of-war in regulating bacterial cell shape.

## Methods

### Bacterial strains and growth conditions

Strains used in this study are listed below. Strains were streaked from −80 °C freezer glycerol stocks onto Nutrient Agar (NA) plates containing the relevant antibiotics and/or inducers and grown overnight at 37 °C. Starter cultures were prepared from a single colony in S750$^{glucose}$ media and grown with orbital agitation at 175 rpm overnight at 37 °C. The next day, overnight cultures were diluted to an OD$_{600}$ of 0.05–0.1 in S750$^{glucose}$ media and grown at 30 °C with orbital agitation at 175 rpm with any required inducer until they reached the appropriate OD$_{600}$. Liquid cultures were grown in flasks with at least a 1:20 culture to flask volume ratio. Overnight cultures were grown in either 2 ml or 5 ml volumes, where day cultures were always grown in 5 ml volumes.

Microscopy was performed at 30 °C. When necessary, antibiotics and inducers were used at the following final concentrations: chloramphenicol 5 μg/ml, spectinomycin 60 μg/ml, erythromycin 1 μg/ml, lincomycin 10 μg/ml, kanamycin 5 μg/ml. S750 media contains 1 X S750 salts, 1 X Metal mix, 10 mM L-Glutamate (Sigma) and 1% carbon source (Glucose or Maltose (VWR)). Metal mix was prepared as a 100 X stock comprising of 2 mM Hydrochloric Acid (HCl) (Honeywell), 190 mM Magnesium Chloride Hexahydrate (Sigma), 65.9 mM Calcium Chloride Dihydrate (Sigma, 4.84 mM Manganese Chloride Tetrahydrate (Fisher), 0.106 mM Zinc Chloride (Sigma), 0.196 mM Thiamine Chloride (Sigma) and 0.470 mM Iron (III) Chloride Hexahydrate (VWR). S750 salts were prepared as a 10 X stock containing 500 mM MOPS (Sigma), 100 mM Ammonium Sulphate (Sigma), Potassium Phosphate Monobasic (VWR) with pH adjusted to 7.0 with Potassium Hydroxide (VWR).

We found that the following considerations were important for reproducible results: (1) S750 must be made up fresh from the relevant stock solutions no more than 1-2 days before the experiments (see S750 preparation protocol in Supplementary Note 2). (2) cells must be cultured in highly aerobic conditions, here in 2–5 ml volumes in 125 ml conical flasks – notable *not* in 16 ml volume test tubes, (3) exponential growth phase cells should be given 10 minutes to recover after being immobilised on the agarose pad before acquiring microscopy data. Deviations from these requirements led to decreased elongasome speed or reduced ratio of motile:immobile elongasomes.

### Strain construction

All strains are derived from the PY79 strain[22]. All experimental strains were constructed with the *Δhag* mutant to disable flagellar motility and reduce cell chaining for VerCINI experiments[12]. *B. subtilis* was transformed in accordance with standard protocols[23]. Oligonucleotides and strains used in this study are detailed in Supplementary Tables 3 and 4.

SM01 (*mreB::mreB-HaloTag, Δhag*) was constructed by transforming bYS40 (Hussain et al.[6]) with gDNA containing *hag::erm* from the *Bacillus* erythromycin (BKE) deletion library (Koo et al.[24]), selecting for transformants on erythromycin and lincomycin. The strain was confirmed by PCR amplification of the *hag* region using oSM01 and oSM02.

SM28 (*mreB::mreB-HaloTag, Δhag, rodA'::Pspac-rodA,*) was produced by transforming SM01 with gDNA from YK2245 (Emami et al.[18]). Transformants were selected on 100 μM IPTG and kanamycin. The strain was confirmed by amplification of the *rodA* region using oSM15 and oSM43. In addition, the strain was streaked on NA plates in the presence and absence of 100 μM IPTG and was found to be IPTG dependent.

SM22 (*mreB::mreB-HaloTag, Δhag, ΔpbpA*) was constructed by transforming SM01 with gDNA containing *pbpA::kan* from the *Bacillus* kanamycin (BKK) deletion library (Koo et al.[24]), selecting for transformants on kanamycin. The strain was confirmed by PCR amplification of the *pbpA* region using oSM19 and oSM20.

SM23 (*mreB::mreB-HaloTag, Δhag, ΔpbpH*) was constructed by transforming SM01 with gDNA containing *pbpH::kan* from the *Bacillus* kanamycin (BKK) deletion library (Koo et al.[24]), selecting for transformants on kanamycin. The strain was confirmed by PCR amplification of the *pbpH* region using oSM27 and oSM28.

SM41 (*mreB::mreB-HaloTag, Δhag, ΔmltG*) was constructed by transforming SM01 with gDNA containing *mltG::kan* from the *Bacillus* kanamycin (BKK) deletion library (Koo et al.[24]), selecting for transformants on kanamycin. The strain was confirmed by PCR amplification of the *mltG* region using oSM78 and oSM79.

MB60 (*mreB::mreB-HaloTag, Δhag, rod'A::Pspac-rodA, amyE::tet pveg-we*) and MB59 (SM28, amyE::tet pveg-empty) were constructed by transforming SM28 with the plasmids MB12 and MB13, respectively. Transformants were selected on 100 μM IPTG and tetracycline. Insertion in *amyE* locus was confirmed by iodine assay. In addition, the strains were streaked on NA plates in the presence and absence of 100 μM IPTG and were found to be IPTG dependent.

MB36 (*mreB::mreB-HaloTag, Δhag, rodA'::Pspac-rodA, murAA::spec pxyl-murAA*) was constructed by transforming SM28 with gDNA from the strain SLA039 (*murAA:: spec pxyl-murAA*). Transformants were selected on 100 μM IPTG, 0.3% xylose and spectomycin. To confirm insertion in *murAA* locus, the strain was streaked on NA in presence and absence of xylose and was found to be xylose dependant. In addition, the strain was streaked on NA plates in the presence and absence of 100 μM IPTG and was found to be IPTG dependent.

MB38 (*PY79, amyE::tet Pveg-msfGFP*) was constructed by transforming PY79 with the plasmid MB10. Transformants were selected on tetracycline. Insertion in *amyE* locus was confirmed by iodine-starch assay.

MB76 (*PY79, amyE::spec Pxyl-msfGFP*) was constructed by transforming PY79 with the gDNA from the strain JWG184. Transformants were selected on spectomycin. Insertion in *amyE* locus was confirmed by iodine-starch assay.

MB37 (*PY79, ΔmreB::neo*) was constructed by transforming PY79 with the gDNA from the strain KS36 (Ωneo3427 ΔmreB). Transformants were selected on kanamycin and 20 mM MgSO$_4$. Absence of *mreB* was confirmed by the inability of the strain to grown without supplemented MgSO$_4$ (20 mM).

MB35 (*PY79, Δmbl::zeo*) was constructed by transforming PY79 with the gDNA from the strain MDS19 (*168ca trpC2 mbl::zeo*). Transformants were selected on zeomycin and 20 mM MgCL$_2$. Absence of *mbl* gene was confirmed by the inability of the strain to grown without supplemented MgSO$_4$ (20 mM).

All used strains are available on request to the authors.

## Plasmid construction

pMB10 (*bla amyE::pveg-msfGFP tet*): the plasmid pCW433 (kindly gifted by Charles Winterhalter) was amplified with MB-F-64 and MB-F-65. The *msfGFP* gene was amplified from gDNA from JWG184 strain with MB-F-62 and MB-F-63 primers (underlined region) and inserted by Gibson assembly. The plasmid has been sequenced to confirm the correct insertion.

pMB12 (*bla amyE::pveg-murAA tet*): the plasmid pCW433 was amplified with MB-F-64 and MB-F-68. The *murAA* gene was amplified from gDNA from BS168 strain with MB-F-66 and MB-F-67 primers (underlined region) and inserted by Gibson assembly. The plasmid has been sequenced to confirm the correct insertion.

pMB13 (*bla amyE::pveg-empty tet*): the plasmid pCW433 was amplified with MB-F-64 and MB-F-65 and closed by Gibson assembly. The plasmid has been sequenced to confirm the correct sequence.

## Growth curves

$OD_{600}$ was measured every 1 h from cultures prepared as described above (Bacterial strains and growth conditions).

## Cell morphology analysis

Cells were prepared for imaging in S750$^{glucose}$ at 30 °C. Once the cultures had reached $OD_{600}$ 0.6 ± 0.1, Nile Red was added to 200 µl of cells to a working concentration of 1 µg/ml, and incubated at growth temperatures for 10 min, prepared on agarose microscope slides as described below and cell morphology images recorded using the microscope described below. To measure cell width, a straight-line ROI was drawn over the short axis of the cell in FIJI and an intensity profile plotted. The intensity plots were exported to MATLAB where the centre of each peak and the distance between them were determined by fitting to a tilted circle model[12]. To measure cell length, a straight-line ROI was drawn from the pole to pole, or pole to septum and the length measured in FIJI.

## Western blotting

For western blot sample preparation, PY79 and SM01 were grown in S750glu while MB35, MB37 and KS60 were grown in LB supplemented with 20 mM MgSO$_4$. Overnight cultures of strains were grown at 37 °C. The following morning, cultures were diluted to $OD_{600}$ ~ 0.05 in 5 ml (125 ml flask) and grown at 37 °C until $OD_{600}$ ~ 0.4. 1 ml of cell culture was harvested by centrifugation and lysed by incubation for 20 min with 200 µl in BugBuster protein extraction reagent supplemented with 1 µl of Benzonase nuclease (Millipore), 0,6 µl of lysozyme (10 mg/ml, Sigma) an EDTA free protease inhibitor cocktail (Roche). Protein contents were quantified by Qubit. Protein extract was heat-denatured for 10 min at 80 °C in NuPAGE LDS Sample Buffer an an equivalent of 2 µg total protein was separated by SDS-PAGE on a NuPAGE 4–12% Tris-Acetate Midi Gel in MOPS buffer (Invitrogen). Protein was transferred to a 0.45 µm PVDF Membrane (Cytiva), and MreB (in-house polyclonal, originally developed by Jeff Errington lab[25]) or SpoOJ (in-house polyclonal, originally developed by Jeff Errington lab[26]) were detected using respective in-house polyclonal antibodies and HRP-conjugated anti-rabbit IgG antibody (Sigma A6154)). Samples were developed using Clarity Western ECL Substrate (Bio-Rad) and imaged using an ImageQuant LAS 4000 mini Biomolecular Imager (GE Healthcare).

## Microscopy

*VerCINI on custom single-molecule microscopes.* Two similar custom single-molecule microscopes were used for experiments. Cells were illuminated with a 561 nm laser (Obis). A 100× TIRF objective (Nikon CFI Apochromat TIRF 100XC Oil) was used. A 200 mm tube lens (Thorlabs TTL200) and Prime BSI sCMOS camera (Teledyne Photometrics) were used for imaging, giving effective image pixel size of 65 nm/pixel. Imaging was done with a custom-built ring-TIRF module operated in ring-HiLO[27] using a pair of galvanometer mirrors

(Thorlabs) spinning at 200 Hz. 8 min time lapses were obtained with 500 ms exposure at a power density of 16.9 W/cm$^2$ at a strobe interval of 6 s unless otherwise stated. Power density was calculated based on 2.5 mW illumination power measured at the sample, over an illumination area of approximately 14,800 µm$^2$. Micro-Manager (v2.0 gamma) was used for microscope control.

*Structured Illumination Microscopy on a Nikon N-SIM.* Cells were illuminated a 561 nm laser (CVI Melles-Griot). A 100× TIRF objective (Nikon CFI Apochromat TIRF 100XC Oil) was used for imaging and an Andor iXon DU897 EMCCD camera was used, with a 2.5× magnifier (Nikon) and standard Nikon tube lens, giving an effective image pixel size of 64 nm/pixel. Cells were illuminated in TIRF-SIM mode, using a 2D-striped pattern. Each SIM image was formed from 9 raw images corresponds to 3 stripe angles and 3 stripe phases. SIM reconstruction was performed using proprietary Nikon software which implements the Gustaffson SIM reconstruction algorithm[28]. Reconstruction was carried out in NIS elements using default settings; Illumination modulation contrast was set to 1.00, high resolution noise suppression was set to 1.00 and out of focus blur suppression was set to 0.05. NS-Elements (v5.42.02) was used for microscope control.

All microscopy was performed on microscopes equipped with incubators to maintain sample and microscope temperature at 30 °C.

## Single-molecule HaloTag labelling with JF-549

At $OD_{600}$ of 0.6 ± 0.1, 500 µl cells were incubated for 15 min with JF-549[13] dissolved in dimethyl sulfoxide (DMSO) to a final concentration of 25 pM at 30 °C with shaking at 175 rpm. Stocks were prepared at concentrations to ensure a working DMSO concentration of <1%. Cells were then washed twice in 500 µl pre-warmed media.

## Sample preparation for VerCINI microscopy

Agarose microholes were formed by pouring molten 6% agarose dissolved in media (typically S750$^{glucose}$ unless otherwise stated) onto a silicon micropillar array as described previously[12]. Patterned agarose was transferred into a Geneframe (Thermo Scientific) mounted on a glass slide, and excess agarose was cut away to ensure sufficient oxygen.

Cultures were concentrated 50-fold and 10 µl was applied to the pad, before centrifugation at 6000 × g for 4 minutes (Eppendorf 5810 centrifuge with MTP/Flex buckets). Pads were then washed with pre warmed media before application of the cover slip (VWR 22 × 22 mm$^2$ Thickness no. 1.5).

VerCINI data were acquired within a 40-min time window after sample preparation, during which no effect on MreB dynamics, for example due to oxygen limitation or phototoxicity, was observed compared to horizontal cell microscopy (Supplementary Fig. 10c, d).

## VerCINI data analysis

**Pre-processing.** Videos were denoised using the ImageJ plugin PureDenoise[29], which is based on wavelet decomposition. For the largest image dataset- 0.5 s frame interval measurements in Fig. 1e – a GPU-accelerated version of PureDenoise was developed and used (http://www.GitHub.com/ZikaiSun/PureGpu). This version also corrects a memory leak bug for large images in the original PureDenoise ImageJ plugin. Performance characterisation of PureDenoiseGPU is shown in Supplementary Tables 5 and 6.

Denoised videos were registered using the ImageJ plugin StackReg[30]. Cropped region of interest (ROI) movies containing single in-focus cells were manually selected and exported for analysis using the publically available scripts (https://github.com/HoldenLab/Ring_Analysis_IJ).

Images were background subtracted and kymographs extracted using a custom fitting model of diffuse out-of-focus cytoplasmic background plus localised protein signal as previously described[12].

**Kymograph analysis of MreB single-molecule dynamics.** In ImageJ, a segmented line ROI was manually traced over each track, with segments indicating manually identified processive or paused subtracks. For each kymograph, an ROI set was saved. Using a custom FIJI plugin, 'Export_XY_Coords.ijm', the coordinates of each point in each track were exported to a '.csv' file. The coordinates of each track were analysed using custom python script 'Track data analysis-Full_.ipynb', to determine MreB filament binding dynamics including bound lifetime, processivity. The required Analysis code is available: https://github.com/HoldenLab/Kymograph-spt-analysis.

Fiji (v1.54) was used for all image analysis in ImageJ. Python (v3.9.13) was used for all data analysis in Python.

## Stroboscopic analysis of photobleaching and MreB binding lifetime

We analysed MreB subunit unbinding lifetime and JF549 photobleaching lifetime using the stroboscopic illumination method of Gebhardt and coworkers[31]. Using a fixed illumination and exposure time of 500 ms, we systematically increased the total time interval between frames, the strobe interval, and measured the apparent lifetime of labelled MreB molecules for each condition (Fig. 1d, e). We calculated the median lifetime of each dataset, with 95% CIs calculated by bootstrapping. We then fit the median lifetime data to Eq. (1),

$$\tau_{obs} = \Delta t \left/ \left( \frac{t_{\exp}}{\tau_{bl}} + \frac{\Delta t}{\tau_{off}} \right) \right. \tag{1}$$

where $\Delta t$ is the strobe interval (the x-axis), $\tau_{obs}$ is the apparent lifetime, $t_{\exp}$ is the fixed 500 ms exposure time, $\tau_{bl}$ is the JF549 photobleaching lifetime and $\tau_{off}$ is the MreB subunit unbinding lifetime. By fitting the data to the median apparent lifetimes, we obtained estimates of median $\tau_{bl}$ and $\tau_{off}$, rather than mean, consistent with the rest of the statistics in the manuscript. We obtained 95% CI estimates for $\tau_{bl}$ and $\tau_{off}$ by bootstrap resampling of the inputs into the stroboscopic fitting equation. MATLAB (R2023a) was used for this analysis.

## Switching rate analysis

We calculated single-molecule switching rates (reversal, pausing, unbinding, initiation) by counting the number of each transition type from immobile or processive states, and dividing by the total duration of all immobile or processive states observed in the dataset[20]. 95% confidence intervals on the switching rate were calculated by bootstrap sampling of individual tracks from the dataset.

## Pveg/Pxyl intensity measurement

MB38 and MB73 were grown in S750glu or S750gly overnight at 37 °C. The following morning, cultures were diluted to OD600 ~ 0.05 in 5 mL (125 mL flask), in S750glu or S750gly with 30 mM xylose, and grown at 37 °C until OD600 ~ 0.5. Once the cultures had reached OD600 0.5 ± 0.1, cells were prepared on agarose microscope slides as described below. James Grimshaw FIJI macro has been used to measure cell average fluorescence (https://github.com/NCL-ImageAnalysis/General_Fiji_Macros).

## Simulations of elongasome dynamics

We implemented stochastic simulations of elongasome tug-of-war using an adaptation of the Müller, Klumpp, and Lipowsky (MKL) model of tug-of-war in eukaryotic cargo transport[17]. The theoretical model and computational parameters are discussed in detail in Supplementary Note 1. A schematic diagram of the elongasome tug-of-war model is presented in Supplementary Fig. 7. MATLAB (R2023a) was used for this analysis. We wrote open source software to perform this simulation (https://github.com/HoldenLab/lipowskiModel).

## Statistics

Experiments were conducted in biological duplicate because variation between clonal bacterial samples was low, as estimated based on small range measured in replicate medians, unless otherwise indicated.

Averages reported were median values unless otherwise indicated. Medians of biological replicates are shown on figures as white-filled circles. 95% confidence interval of the median, or of the difference of medians, was estimated by bootstrapping. Interquartile range was indicated by IQR. Thick error bar lines in violin plots indicate interquartile range, thin lines indicate adjacent vales. Because variability between single-molecules was far greater than the sample-to-sample variation, estimates of uncertainty (95% Cis, IQRs, etc.) were based on the single-molecule datapoints. Sample size, indicating number of tracks/ track segments, technical and biological replicates, as appropriate, is presented for each dataset in Supplementary Table 7.

We checked that single cell/ single molecule variability was the relevant point of statistical comparison and that variation between independent biological replicates was low based on at least two biological replicates, which is presented for all figures and analyses.

Effect size estimates and confidence intervals[32] were calculated based on difference of medians using DABEST (Data Analysis with Bootstrap Coupled Estimation[33]) unless stated otherwise, difference of medians using custom bootstrapping scripts (for the switching rate kinetics data), or difference of means and margin of error calculations[32] (for small sample size calculations, ie growth rate and cell diameter). All effect sizes are listed in Supplementary Table 2. In comparison to null hypothesis significance testing (NHST) analysis, which only reports on whether a difference may exist between two populations, effect size analysis addresses one of the flaws of NHST by allowing analysis of not only whether a difference may exist but also the magnitude of that difference[32], which is often of critical importance for interpreting the biological significance of a result. For comparison to NHST analysis, a median/ mean observed difference whose 95% CIs are both greater or less than zero this would indicate rejection of the null hypothesis that the two population medians/ means are equal at $p = 0.05$ significance level.

For estimates of the percentage of the population in a specific state (eg percentage of motile tracks), uncertainty is reported to the full data range (*Range*) of all biological replicates.

## Reporting summary

Further information on research design is available in the Nature Portfolio Reporting Summary linked to this article.

## Data availability

Source data for all figures presented in the paper and Supplementary Information, as well as representative raw video data, are available at https://doi.org/10.6084/m9.figshare.25922524.

## Code availability

Open source software for image analysis of VerCINI data, described in Ref. 12 https://github.com/HoldenLab/VerciniAnalysisJ, https://github.com/HoldenLab/ring-fitting2[34], Open source software for kymograph analysis available on the Holden Lab GitHub page: https://github.com/stu-middlemiss/Kymograph-spt-analysis/tree/Kymo[35]. Open source PureDenoise-GPU denoising software: http://www.GitHub.com/ZikaiSun/PureGpu[36]. Open source software for the tug-of-war simulations: https://github.com/HoldenLab/lipowskiModel[37]. Open source software for bacterial image analysis: https://github.com/NCL-ImageAnalysis/General_Fiji_Macros[38].

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

## Acknowledgements

We thank Yoshi Kawai, Jeff Errington, Richard Daniel (Newcastle) and Ethan Garner (Harvard) for strains. We thank Charles Winterhalter (Newcastle) for plasmids. We thank Cees Dekker (TU Delft, Netherlands) for nanofabricated VerCINI wafers and Luke Lavis (Janelia Farm, USA) for Janelia Fluor dyes. We also thank Waldemar Vollmer (Queensland) for insightful discussions. S.H. acknowledges funding support by a Wellcome Trust & Royal Society Sir Henry Dale Fellowship grant number 206670/Z/17/Z and a UK Biotechnology and Biological Sciences Research Council (BBSRC) grant BB/X001482/1. S.M. was supported BBSRC doctoral studentship (BB/M011186/1). H.S. acknowledges funding from BBSRC grants BB/S00257X/1 and BB/X001512/1. T.B. acknowledges funding support from the Yushan Fellow Program (MOE, Taiwan).

## Author contributions

S.M., M.B. and J.G. constructed and characterised bacterial strains. S.M., M.B., A.M., D.M.R. and S.H. performed experiments. S.M., A.M., D.M.R.,

H.S. and S.H. analysed data. Z.S. and T.B. developed PureDenoiseGPU software. K.W. tested PureDenoiseGPU software. S.H. performed simulations. J.E. designed and built one of the custom microscopes used in the study. S.H., S.M and H.S. designed experiments. S.H. and H.S. directed the research. S.H. and S.M. wrote the manuscript with input from all authors.

## Competing interests

The authors declare no competing interests.
