## [Peer Review File · Nature Communications]

Molecular motor tug-of-war regulates elongasome cell wall synthesis dynamics in *Bacillus subtilis*Reviewer #1 (Remarks to the Author):

This paper presents experiments that allow to track the motion of MreB filaments driven by peptidoglycan synthesis for unprecedented long times. This is made possible by a clever imaging methods, single molecule tracking in a vertically oriented cell (VerCINI). The motion is found to be typically bidirectional, a fact that is interpreted with a tug-of-war model which is then tested by additional analyses.

This is a beautiful experiment that provides a very detailed picture of the dynamics, together with an original interpretation, supported by simulations. Overall, this is a great study, and I am strongly in favor of its publication.

I only have a few minor comments:

1. related to Fig. 1: Does processivity relate to the circumference of the bacterium in some way? Some of the long processive tracks exceed the circumference. Are these different from short tracks in some way?
2. Likewise, do pauses and reversal occur randomly or are there preferred positions?
3. What is plotted as a displacement in fig. 1g, total distance moved (forward+backward) or net displacement (forward-backward)?
4. line 147: what is meant with „substrates“? Typo „substates“?
5. Fig. 3f: Why does the processivity saturate? Is this real or do to limited sampling of long tracks?
6. The statement that tug-of-war has only been seen for eukaryotic motors (p.11, line 337) is not entirely true. Tug-of-war has also been proposed for bacterial motility driven by type-IV pili (Marathe et al. 2014).
7. A short summary of the simulations could be included in Methods.

Reviewer #2 (Remarks to the Author):

This is an impressive piece of work, in which the authors have leveraged single molecule tracking and vertical cell imaging to study the dynamics of MreB in unprecedented detail – in terms of tracking filaments over completely different scales of both length and time as has been done until now. The observed frequent reversal of filament movement is spectacular and requires new models to interpret MreB dynamics, which the authors propose.

There are two things I would like to see explained better. First – what exactly are we looking at in these experiments? I think a more detailed (supplemental) description of how to interpret the kymographs would greatly help readers understand the paper and avoid confusion. Second, I am not completely convinced by the proposed model – as I interpret it, the authors consider that the supply of precursor for the elongasome is not limiting in their experiments. A recent paper from the Garner lab suggests that it is limiting. Limiting substrate may also explain the observed reversals of direction etc. The authors should discuss these discrepancies and, ideally, provide data that supports the notion that the concentration of elongasome complexes, and not the concentration of precursor, is limiting in their experimental conditions.

- 1) The authors compare the tug of war with eukaryotic models. In the eukaryotic model described in the paper cited by the authors, tug of war is mediated by motors that are moving along a pre-existing microtubule filament towards one of the ends of the microtubule. Is this conceptually

similar to the MreB filament? If RodA binds to the ends of the filament, isn't filament elongation following RodA movement? And once a MreB molecule is incorporated in the filament is it stationary (movement that is observed only reflecting the incorporation of new subunits at the RodA tugging end of the filament, ie as in treadmilling) or is RodA actually dragging a filament along meaning that the incorporated individual MreB molecules track along the membrane? Basically – what exactly is meant by filament motility?

It is possible that these concepts are very clear cut for biophysicists, but for (cell) biologists, such as myself, it is quite difficult to distinguish these possible scenario's or to know which one the authors actually mean. I would really appreciate some (supplementary) figures that clarify how to interpret the data:

How many fluorescent MreB molecules are in a filament? Can the authors provide a cartoon of a sequence of events to make very clear what the observed movement represents?

What is the (average) length of MreB filaments?

Does 'reversal' mean the same filament now being pulled the other way, or that incorporation of new material is now taking place in the other direction?

2) L 157-158 and Supplementary Fig 3. The authors find that the speed of MreB is not dependent on growth rate. This is in line with findings in a recent paper from the Garner lab (Sun et al., 2023), but contrasting with results published by other labs (Billaudeau et al., 2017, Zielińska et al., 2020). The Garner lab paper did observe a difference in filament density depending on growth rate. Did the authors observe this as well?

More importantly, the paper from the Garner lab provided evidence for a model where cell growth is regulated by the availability of Lipid II, the substrate for RodA.

The tug of war model presented here, if I understand it correctly, depends on active (line 199) synthesis complexes acting in opposite directions on a filament. By testing this model via upregulation of RodA, the authors appear to assume that there is no limit to the supply of substrate Lipid II in their experimental conditions. This is in contrast with the findings from the Garner lab. Have the authors considered the possibility that upregulation of RodA results in an increased number of elongasome complexes that pause on and/or unbind from the MreB filament more frequently because of lack of substrate – which would result in either the pausing or reversal of MreB dynamics if another elongasome complex on the filament becomes active? The widening of cells upon RodA overexpression – which the authors consider the principal factor controlling the concentration of active synthesis complexes (line 217-8) - is dissimilar to the narrowing of cells described by the Garner lab when PG synthesis is accelerated via upregulation of precursor synthesis. It is also counterintuitive to the narrowing of cells observed when elongasome activity is (supposedly) stimulated by overexpressing MreB instead of RodA (Dion et al., 2019).

In my view, these considerations need to be discussed – does the concentration of RodA or LipidII (or MreB, or all of these) control the amount of active synthesis complexes? Ideally the authors should provide additional evidence showing that: either Lipid II is not limiting in their experimental conditions; or that conditions in which Lipid II production is upregulated do not alter their findings that increased RodA results in increased reversals.

Minor comments:

L 19 "frequent" – frequently?

L 42-44 "primarily performed". This is not entirely clear – the authors base the statement on two studies - one in *B. subtilis*, where either system (rod/aPBP) can support rod-shape and the other in *E. coli*, where aPBPs seem mostly involved in repair and are actively regulated by OM proteins that sense PG 'gaps'. However, the PG of *B. subtilis* is quite different from that of *E. coli*, and it is also not known how much 'mature' PG is actually the result of repair events, eg required to allow expansion of the cell wall. So – is there any quantitative data that underpins the statement "primarily performed"? If not, the statement should be rephrased.

L256 "assume that that..."

L392 Can the authors briefly explain why they used Dhag strains?

L414 The authors stress the need for highly aerobic conditions (L380-6) for microscopy, yet all

growth curves are determined in microtiter plates which have low aeration. It is likely that the cells of which the growth rates were determined (supplemental figs 3 and 8) were growing slower than the cells that were imaged.

BILLAUDEAU, C., CHASTANET, A., YAO, Z., CORNILLEAU, C., MIROUZE, N., FROMION, V. & CARBALLIDO-LOPEZ, R. 2017. Contrasting mechanisms of growth in two model rod-shaped bacteria. *Nat Commun*, 8, 15370.
DION, M. F., KAPOOR, M., SUN, Y., WILSON, S., RYAN, J., VIGOUROUX, A., VAN TEEFFELLEN, S., OLDENBOURG, R. & GARNER, E. C. 2019. *Bacillus subtilis* cell diameter is determined by the opposing actions of two distinct cell wall synthetic systems. *Nat Microbiol*.
SUN, Y., HÜRLIMANN, S. & GARNER, E. 2023. Growth rate is modulated by monitoring cell wall precursors in *Bacillus subtilis*. *Nature Microbiology*.
ZIELIŃSKA, A., SAVIETTO, A., DE SOUSA BORGES, A., MARTINEZ, D., BERBON, M., ROELOFSEN, J. R., HARTMAN, A. M., DE BOER, R., VAN DER KLEI, I. J., HIRSCH, A. K. H., HABENSTEIN, B., BRAMKAMP, M. & SCHEFFERS, D.-J. 2020. Flotillin-mediated membrane fluidity controls peptidoglycan synthesis and MreB movement. *eLife*, 9, e57179.

Reviewer #3 (Remarks to the Author):

There is much to like in this new study of the dynamics of the *B. subtilis* elongasome. The authors considerably extend previous studies done by TIRF microscopy by orienting cells vertically to allow detection of movement over longer distances and times than was possible previously. This single-molecule "VerCINI" approach reveals long processive movements of MreB that frequently reverse direction, pause, and restart. Notably, the reversals and pauses were much more frequent than observed before, likely because of limitations of the TIRFm approach. Additional measurements by stroboscopic illumination reveal a relatively long lifetime of MreB subunits within filaments, and allow measurements of photobleaching. The smVerCINI results indicate relatively long lifetimes of processive and paused motility states. The processivity at constant velocity is now determined to correspond approximately to half the diameter of the cell. Interestingly, these movement parameters are not strongly changed in cells grown under different conditions. A second series of experiments shows that modulating RodA glycosyltransferase amounts changes the bidirectional motility and processivity, with speed and processivity increasing at lower RodA amounts. These results are interpreted in terms of "tug-of-war" models in which multiple elongasome complexes bind to MreB filaments. Another implication of this work is that RodA amount is limiting in *B. subtilis* cells growing under these conditions, whereas the cognate Class B PBPs seem to be in excess. In the last part of this paper, the authors model different versions of the "tug-of-war" model. This modeling favors an end-binding model over a multiple binding model. Remarkably, the end-binding model can account for the observation that cell diameters increase at low and high levels of RodA induction, which has not been accounted for before.

This is a succinct, innovative paper that provides new insights into the elongasome dynamics of the experimental system used. The paper is convincing and the data are convincing as far as it goes. The paper will be of considerable interest and impact to the field. Significantly, the paper critically revives the "tug-of-war" model, which seems to fit with this new data. There are a couple points about the cell physiology that need to be mentioned and clarified in the paper. In addition, to help readers not in this area, the authors might include some definitions of some parameters and brief heuristic explanations of some of the methods. Some suggestions and comments follow.

1. The paper strongly shows the importance of cellular concentrations for elongasome components, in this case RodA, on outcomes of these dynamics experiments. The motion determinations are of a previously reported MreB-HT fusion reporter. More should be said about this MreB-HT fusion here. What was its cellular concentration compared to the WT MreB protein? Was it fully functional in the absence of the other MreB homologs? If the MreB-HT amount or activity is not WT, how would this affect the interpretation of the experiments?

2. Along a similar line, what is known about the relative stoichiometry of RodA compared to PBP2A and PBPH in *B. subtilis* cells? It seems that this must be known and should be discussed as context. To further test the idea that RodA is limiting, does overexpression of PBP2A and/or PBPH affect MreB velocity? Last, were the extent of RodA under- and overexpression (e.g., Fig. 2f) determined by western blotting? Does the level of RodA amount in the controlled strain correspond to the WT level when the WT speed and processivity are observed.

3. Line 106 and Fig 1d-e. A bit more might be explained about the the stroboscopic illumination experiments.

4. Line 137. Add "(Fig. 1f)" at end of the line.

5. Fig. 1c is very hard to read because of its size and color, and Fig. 1i is not drawn properly (see Supplemental Fig. 3).

6. Some definitions or brief explanations for some terms would help improve clarity, including processive subtracks, circumferential track displacement, elongasome displacement, track density, processive and static track lifetimes.

7. Fig. 3. In the end-binding model, are both synthases actively synthesizing in opposite directions or is only one synthase actively synthesizing and the other binding the other end, as depicted? It would seem that both synthases would be active in opposite directions to get a tug-of-war effect. Please clarify.

8. Along the same line, can synthase complexes (e.g., RodA-sfGFP) be detected at the ends of MreB-HT(red) filaments by SIM-TIRF in two-color experiments? This seems like a relatively simple experiment that would provide direct and compelling evidence for the end-binding model, and would indicate if unlimited binding does occur at all.

9. Fig. 4b. The conclusion is no difference in track density, but the graph seems to indicate a difference. What do the statistics (P values) say? Likewise, in Fig. 4a and in other comparisons, the trends and distributions look very nice based on large numbers of determinations, but are the differences statistically different. Please clarify this point here and elsewhere.

10. Line 331. Please consider qualifying a bit. More needs to be done to definitively support the "tug-of-war" model (see point 8, above), but the case made here is strong for support of this model.

11. Supplemental Fig. 1. How do the growth parameters compare to the WT strain that lacks MreB-HT? There is no comparison here (please see point 1, above).

REVIEWER COMMENTS

We thank all the reviewers for their constructive comments which we comprehensively address below. Text and figure legend changes referenced below have been highlighted in the main text and SI.

Reviewer #1 (Remarks to the Author):

This paper presents experiments that allow to track the motion of MreB filaments driven by peptidoglycan synthesis for unprecedented long times. This is made possible by a clever imaging methods, single molecule tracking in a vertically oriented cell (VerCINI). The motion is found to be typically bidirectional, a fact that is interpreted with a tug-of-war model which is then tested by additional analyses.

This is a beautiful experiment that provides a very detailed picture of the dynamics, together with an original interpretation, supported by simulations. Overall, this is a great study, and I am strongly in favor of its publication.

I only have a few minor comments:

1. related to Fig. 1: Does processivity relate to the circumference of the bacterium in some way? Some of the long processive tracks exceed the circumference. Are these different from short tracks in some way?

To clarify what processivity means and how it relates to cell circumference we have added Supplementary figure 2, which is an extension of figure 1 to show how kymographs are plotted, and how dynamics including processivity are quantified the kymographs. We have now added a reference to SI Figure 2 (Line 106-107):

“We analysed the data using kymographs (see schematic Supplementary Figure 2, also definitions SI Table 1).”

In addition, we have added the following text in the supplementary information (SI Lines 44-55) for further explanation.:

' To analyse MreB tracking data, kymographs were produced by plotting a ring around the cell circumference based on signal intensity (SI Fig. 2c). The example kymographs (SI Fig. 2c) show MreB displacement around the cell circumference on the x-axis, over time in the y-axis. A diagonal line on the kymograph signifies an MreB molecule moving around the cell circumference and there is a change in both displacement and time, whereas a vertical line shows a static molecule, as there is no displacement around the cell circumference over time.

As we found MreB to be highly processive, with processivity often exceeding one circumference of the cell (360°), it was necessary to plot kymographs over multiple circumferences by repeating the plot multiple times in the x-axis. This is annotated with a dashed yellow line to show each 360° cell circumference. During image analysis, all kymographs were plotted and analysed with 6 repeats of the cell circumference in the x-axis which ensured that no tracks were falsely truncated. The extent at which it was necessary to plot kymographs multiple times around the cell circumference is evident when processivity is plotted as displacement in degrees around the cell circumference (SI Fig. 3).'

Further, we have added Supplementary Table 1, which lists all of the terms we use to describe elongasome dynamics with definitions and explanations of how they are measured.

To clarify how elongasome processivity relates to distance travelled around the cell circumference, we have added Supplementary Figure 3, which shows elongasome processivity plotted in degrees around the cell circumference (referred to in line 174).

Some of the longer tracks indeed exceed the circumference, i.e. the processivity is $> 360^\circ$. In this case, it means that an elongasome synthesis event completes more than one full circle around the cell circumference. In fact, the longest synthesis events moved 3 entire turns around the cell circumference! We do not have any evidence that there are differences in short vs long tracks except for their duration.

2. Likewise, do pauses and reversal occur randomly or are there preferred positions?

This is an interesting point that we also considered. This could speak to for example influence of the underlying cell wall on elongasome dynamics. If preferred sites for bidirectional motility were to have a major role in regulating elongasome dynamics, we would expect these features to be frequent and fairly obvious even by eye. We looked in depth at the kymographs and could not find any points at which reversals and pauses occurred at similar positions, beyond the rare occasions that one would expect from coincidence. Further, if the underlying cell wall template was the primary regulator of elongasome processivity, we would not expect such a large increase in processivity that we observe when we deplete RodA. It is possible that preferred positions could exist to some small degree but if so they must play a minor role in elongasome dynamics regulation.

We have now mentioned this point in the results section (Lines 165-168)

‘The positions of elongasome reversals and pauses did not show any obvious pattern, suggesting that such switching dynamics are caused by intrinsic factors within the elongasome, rather than local heterogeneity in the cell wall template.’

3. What is plotted as a displacement in fig. 1g, total distance moved (forward+backward) or net displacement (forward-backward)?

Total displacement is gross (forward + backward) displacement and processive subtrack displacement is displacement in one direction before any pause or reversal. This is clarified in Supplementary Table 1 and the legend of Figure 1 (Lines 142-144)

‘Total tracks refer to the whole observed elongasome trajectory, whereas subtracks refer to regions within a track whose motion type remains constant, ie constant processive motion or pausing.’

4. line 147: what is meant with „substrates“? Typo „substates“?

Thank you, this has been amended.

5. Fig. 3f: Why does the processivity saturate? Is this real or due to limited sampling of long tracks?

We have addressed this question in the results section (Lines 329-334):

‘Consistent with previous analyses of the MKL model for eukaryotic motor tug of war [17], at high synthase concentration the unlimited binding model predicts almost indefinitely processive elongasomes - limited here only by the length of the simulations. This is because individual elongasomes are very unlikely to win a tug-of-war against large numbers of oppositely bound active synthases. This runaway-motor scenario is inconsistent with experimental results.’

6. The statement that tug-of-war has only been seen for eukaryotic motors (p.11, line 337) is not entirely true. Tug-of-war has also been proposed for bacterial motility driven by type-IV pili (Marathe et al. 2014).

This is a helpful point. We were not aware of this study. After reading the study we believe our statement is still valid but needs extra clarity. The tug-of-war observed in the mentioned study is on a cellular level of multiple molecular assemblies acting against one another. In our model, the tug-of-war is unique in that it occurs at a molecular level, within one assembly – like eukaryotic motor tug of war. We removed the phrase ‘previously thought to be a phenomenon exclusive to eukaryotic molecular motors’ and added ‘This phenomenon, previously thought to be exclusive to eukaryotes is the first of its kind described on a molecular level in bacteria.’ (Lines 396-398)

7. A short summary of the simulations could be included in Methods.

We have added detail of the simulations in the methods section (Lines 615-619):

‘We implemented stochastic simulations of elongasome tug-of-war using an adaptation of the Müller, Klumpp, and Lipowsky (MKL) model of tug-of-war in eukaryotic cargo transport [4]. The theoretical model and computational parameters are discussed in detail in Supplementary Note 1, a schematic diagram of the model is presented in Supplementary Figure 10’

Reviewer #2 (Remarks to the Author):

This is an impressive piece of work, in which the authors have leveraged single molecule tracking and vertical cell imaging to study the dynamics of MreB in unprecedented detail – in terms of tracking filaments over completely different scales of both length and time as has been done until now. The observed frequent reversal of filament movement is spectacular and requires new models to interpret MreB dynamics, which the authors propose.

There are two things I would like to see explained better.

First – what exactly are we looking at in these experiments? I think a more detailed (supplemental) description of how to interpret the kymographs would greatly help readers understand the paper and avoid confusion.

This is a fair comment. We have now addressed this - see reviewer 1 point 1 response.

Second, I am not completely convinced by the proposed model – as I interpret it, the authors consider that the supply of precursor for the elongasome is not limiting in their experiments. A recent paper from the Garner lab suggests that it is limiting. Limiting substrate may also explain the observed reversals of direction etc. The authors should discuss these discrepancies and, ideally, provide data that supports the notion that the concentration of elongasome complexes, and not the concentration of precursor, is limiting in their experimental conditions.

The proposal that a reduction in amount of available lipid II substrate upon *rodA* overexpression could explain our observed data is interesting, however, it does not fit with our data, nor with additional control experiments that we have now performed and included in the manuscript.

If lipid II levels dropped substantially upon *rodA* overexpression as a result of there being more enzymes to use up the substrate, the plausible consequence would be that elongasome complexes (1) pause/ terminate synthesis more frequently and/ or (2) unbind/ disassemble more frequently. We observe neither of these in our data (Figure 2D). Furthermore, substrate limitation cannot plausibly explain an increased rate of elongasome reversals, as this requires the same or higher levels of elongasome activity, not less.

We also note that in the medium to high tug-of-war regime where we see the most interesting changes in both bidirectional motility dynamics and the possibly tug-of-war dependent cell widening, we do not see any changes in MreB filament density which could arise from the PrkC regulation mechanism identified by Sun and coworkers [1] (Figure 4B, 10-1000uM IPTG induction of *rodA* expression).

Therefore, we think that the reviewer's alternative model is not consistent with our data.

We nevertheless performed additional experiments to further investigate this point. The key relevant proposals of the Sun and coworkers study that the reviewer refers is that lipid II concentrations are rate limiting for cell growth and that increased lipid II levels can substantially increase cell growth rate. We note that these conclusions rest on the observation in a single nutrient limited minimal media condition where both lipid II levels and cell growth rate were observed to increase upon *murAA* overexpression.

We therefore tested whether overexpression of *murAA* and thereby altering lipid II concentration could play a role in determining elongasome dynamics. To match the conditions of the Sun & Garner study where the lipid II and growth rate increase upon *murAA* expression was observed [1], these experiments were performed with RodA expressed from its native locus, using the PY79 strain and S750 minimal medium. In particular, this allowed us to perform experiments using conditions where *murAA* overexpression had previously been shown to increase cellular levels of lipid II [1].

We used a strong, constitutive promoter to overexpress *murAA* in order to reproduce data reported by Sun and coworkers, where both lipid II levels and growth rate was found to increase upon *murAA* overexpression [1]. However, we could not replicate the previously reported growth phenotype and rather found that *murAA* overexpression did not increase growth rate in the S750 media tested, with either glycerol or glucose as the carbon source (SI Figure 8a-b).

In addition, we could not detect any effect of *murAA* overexpression on cell morphology, elongasome speed or elongasome processivity in S750 glucose (SI Figure 8c-e). We did not measure elongasome dynamics in S750 glycerol as we found that *Bacillus subtilis* cells rapidly stopped growing during agarose pad-based microscopy experiments performed in that medium.

Based on these experiments, we conclude that lipid II levels are unlikely to play a major role in regulating elongasome bidirectional motility. Our results also question the proposal by Sun and coworkers that cells store excess cell elongation capacity during growth in minimal media.

We added Supplementary Figure 8 to show our efforts to overproduce lipid II. We added the following to the main text (Lines 257-271):

“We also considered whether cellular levels of lipid II could play a role in regulating elongasome dynamics. Overexpressing *murAA*, an enzyme at the beginning of the lipid II synthesis pathway has previously been observed to lead to increased lipid II cellular levels and altered cell growth rate in minimal medium [20]. However, upon overexpression of *murAA* in minimal medium we did not observe changes in cell growth rate and morphology, elongasome processivity or elongasome speed relative to native *murAA* expression levels (Supplementary Figure 8, Supplementary Table 6), suggesting that lipid II levels do not rate limit cell growth under our experimental conditions. Additionally, if lipid II levels dropped substantially upon *rodA* overexpression because of the presence of more enzymes to consume the substrate, the likely consequences for elongasome dynamics would be that elongasome complexes (i) pause or terminate synthesis more frequently and/ or (ii) unbind or disassemble more frequently. We observe neither of these effects in our data (Figure 2D). At last, substrate limitation cannot plausibly explain the observed increased rate of elongasome reversals upon *rodA* overexpression, as this requires the same or higher levels of elongasome activity, not less. These data indicate that cellular lipid II levels are unlikely to play a major role in regulating elongasome bidirectional motility.”

1) The authors compare the tug of war with eukaryotic models. In the eukaryotic model described in the paper cited by the authors, tug of war is mediated by motors that are moving along a pre-existing microtubule filament towards one of the ends of the microtubule. Is this conceptually similar to the MreB filament? If RodA binds to the ends of the filament, isn't filament elongation following RodA movement? And once a MreB molecule is incorporated in the filament is it stationary (movement that is observed only reflecting the incorporation of new subunits at the RodA tugging end of the filament, ie as in treadmilling) or is RodA actually dragging a filament along meaning that

the incorporated individual MreB molecules track along the membrane? Basically – what exactly is meant by filament motility?

This is a fair question that should be clarified. The whole MreB filament moves as one unit transported by active elongasome synthesis complexes and, hence, does not treadmill. Furthermore, the subunit turnover has been found to be negligible by FRAP experiments [2]. How our model relates to the eukaryotic model is actually a little bit “on its head”. Here the MreB filament corresponds to the cargo which is transported by the elongasome synthesis complexes and the existing cell wall corresponds to the track. In the eukaryotic model, in contrast, the cargo is a vesicle or organelle and the track is the cytoskeleton.

To explain this more clearly we have added Supplementary figure 10. This figure describes the differences and similarities between our model, and the eukaryotic MLK model and the molecular mechanism we propose to drive tug-of-war switching dynamics. Also, in the results section, we have described this in more detail (Lines 205-214).

‘In the eukaryotic model, cargo such as lipid droplets is transported bidirectionally along microtubules by competing molecular motors which constantly attempt to drag the cargo in opposite directions [17]. In the elongasome tug-of-war model, the molecular motors (peptidoglycan synthases) move processively along the cell wall track, which is analogous to the microtubule track in the eukaryotic model. The cargo in the case of the elongasome is the MreB filament. PG synthesis results in processive motion of both the elongasome synthases and the MreB filament relative to the cell wall (Fig. 3a, SI Fig. 10). Motor competition in the case of the elongasome corresponds to attempts by synthesis complexes to initiate cell wall synthesis in the opposite direction to the current direction of elongasome cell wall synthesis (Fig. 3a, SI Fig. 10).’

It is possible that these concepts are very clear cut for biophysicists, but for (cell) biologists, such as myself, it is quite difficult to distinguish these possible scenarios or to know which one the authors actually mean. I would really appreciate some (supplementary) figures that clarify how to interpret the data:

How many fluorescent MreB molecules are in a filament? Can the authors provide a cartoon of a sequence of events to make very clear what the observed movement represents?
What is the (average) length of MreB filaments?

We have added a cartoon in Supplementary Figure 2 to show single molecule labelling of MreB monomers within a filament and how data are interpreted. Also, in the introduction, we have added (Lines 51-53).

‘Each double filament is around 170 nm long [5] and monomers are around 5 nm long [4], suggesting the average MreB double filament consists of around 68 subunits.’

As in the previous comment, we have added in Supplementary Figure 10 to show the molecular mechanism of tug-of-war switching dynamics that we propose.

Does ‘reversal’ mean the same filament now being pulled the other way, or that incorporation of new material is now taking place in the other direction?

We cannot yet state this unequivocally but reversal most likely corresponds to both reversal of the filament and incorporation of new material in the opposite direction. I.e. upon reversal it is likely that the current synthesis event is terminated and PG synthesis is initiated in the opposite direction. We hypothesise that the whole MreB filament then moves in the opposite direction, so the leading end of the filament becomes the lagging end.

We have added (Lines 153-157)

‘MreB filaments frequently switch between motile and paused states, and motile MreB filaments frequently switch direction (reversal), which likely corresponds to initiation of PG synthesis in the opposite direction ’

and included a schematic in Supplementary Figure 10 to describe this.

2) L 157-158 and Supplementary Fig 3. The authors find that the speed of MreB is not dependent on growth rate. This is in line with findings in a recent paper from the Garner lab (Sun et al., 2023), but contrasting with results published by other labs (Billaudeau et al., 2017, Zielińska et al., 2020). The Garner lab paper did observe a difference in filament density depending on growth rate. Did the authors observe this as well?

We went back and read again the discussion in the recent Garner lab paper (Sun et al, Nat Micro 2023) where they work with the Carballido-Lopez lab to try to get to the bottom of the discrepancies between the two labs results (Extended data Figure 1).

After their collaborative investigations, we think the discrepancy between the results of the two labs is more minor than it previously appeared. Both Carballido Lopez and Garner groups find small but fairly reproducible variations in speed across various different minimal media growth conditions – we see the same trend (SI Figure 3B). However, for growth rate in LB, Carballido Lopez measure MreB speed of ~70 nm/s whereas Garner measurement of MreB speed of ~40 nm/s. LB is a rich medium where the concentration of the components can vary substantially – I have previously had conversations with other *Bacillus subtilis* community colleagues about variation in LB recipes/ suppliers leading to different growth rates so I am not surprised that this is where they saw substantial variation.

Precisely because the switch to rich media is both variable depending on media composition, and because it causes such substantial and varied changes to overall cell physiology, we took the more controlled approach of only changing the carbon source in the same defined S750 medium. In this case we saw small but non-negligible changes in MreB speed, consistent with the results of both the labs mentioned (SI Figure 3).

We did not measure the MreB filament density for different growth media – this would entail additional TIRF SIM measurements and would not affect the conclusions of our study.

We have added ‘The speeds we observed were similar to those measured previously [3,4] (Lines 176-177).

More importantly, the paper from the Garner lab provided evidence for a model where cell growth is regulated by the availability of Lipid II, the substrate for RodA.

The tug of war model presented here, if I understand it correctly, depends on active (line 199) synthesis complexes acting in opposite directions on a filament. By testing this model via upregulation of RodA, the authors appear to assume that there is no limit to the supply of substrate Lipid II in their experimental conditions. This is in contrast with the findings from the Garner lab. Have the authors considered the possibility that upregulation of RodA results in an increased number of elongasome complexes that pause on and/or unbind from the MreB filament more frequently because of lack of substrate – which would result in either the pausing or reversal of MreB dynamics if another elongasome complex on the filament becomes active?

Please see response to earlier comment where we address the point regarding lipid II levels.

Please see also responses to comments 2 and 10 by reviewer 3.

The widening of cells upon RodA overexpression – which the authors consider the principal factor controlling the concentration of active synthesis complexes (line 217-8) - is dissimilar to the narrowing of cells described by the Garner lab when PG synthesis is accelerated via upregulation of precursor synthesis. It is also counterintuitive to the narrowing of cells observed when elongasome activity is (supposedly) stimulated by overexpressing MreB instead of RodA (Dion et al., 2019).

We should clarify that the cell shape phenotype upon *rodA* under- and over-expression compared to wild type levels – cell widening in either case – was previously observed in the Dion et al 2019 paper and our phenotype matches theirs – the only difference is a slightly different strain background and a different growth medium. Therefore, the phenotype is fully consistent with the cell shape phenotype described by the Garner lab.

The point where ours and their papers differ regarding this phenotype, is that we propose a new mechanistic model for the cause of cell widening upon *rodA* overexpression, which is tug-of-war mediated reduction in elongasome processivity, vs their hypothesis of non-elongasome associated peptidoglycan synthesis by RodA.

In my view, these considerations need to be discussed – does the concentration of RodA or LipidII (or MreB, or all of these) control the amount of active synthesis complexes? Ideally the authors should provide additional evidence showing that: either Lipid II is not limiting in their experimental conditions; or that conditions in which Lipid II productions is upregulated do not alter their findings that increased RodA results in increased reversals.

Please see response to earlier comment where we address the point regarding lipid II levels.

Regarding effect of RodA concentration on density of active synthesis complexes: this point is already investigated in Figure 4B. Depletion of RodA leads to a drop in the density of active synthesis complexes, but overexpression of RodA does not change the density of active synthesis complexes. As overexpression of RodA still changes elongasome dynamics very substantially, this cannot be mediated by changes in the density of synthesis complexes.

Minor comments:

L 19 “frequent” – frequently?

Amended, thank you.

L 42-44 “primarily performed”. This is not entirely clear – the authors base the statement on two studies - one in *B. subtilis*, where either system (rod/aPBP) can support rod-shape and the other in *E. coli*, where aPBPs seem mostly involved in repair and are actively regulated by OM proteins that sense PG ‘gaps’. However, the PG of *B. subtilis* is quite different from that of *E. coli*, and it is also not known how much ‘mature’ PG is actually the result of repair events, eg required to allow expansion of the cell wall. So – is there any quantitative data that underpins the statement “primarily performed”? If not, the statement should be rephrased.

We have changed the statement to read ‘This is performed by class A penicillin binding proteins (PBPs) and a highly conserved protein complex, the elongasome, which inserts long peptidoglycan strands circumferentially around the cell, giving rise to a rod-shaped cell morphology [5,6].’ (Lines 13-14 and 43-45).

L256 “assume that that...”

Corrected, thank you.

L392 Can the authors briefly explain why they used Dhag strains?

The *hag* gene is deleted to knockout the flagella, ensuring cells remain stationary while in the VerCINI setup. We also found this gave the additional benefit of reducing the chaining phenotype of *B. subtilis*, which was useful for increasing the concentration of cells loaded into the VerCINI microhole arrays. This is described in detail in Whitley et al, Nature Protocols 2022.

This is mentioned in the methods section (Lines 448-449).

L414 The authors stress the need for highly aerobic conditions (L380-6) for microscopy, yet all growth curves are determined in microtiter plates which have low aeration. It is likely that the cells of which the growth rates were determined (supplemental figs 3 and 8) were growing slower than the cells that were imaged.

We agree that this is important. We have removed data taken using the plate reader and only included growth data that were taken from cultures grown in the same way as the cultures used for all elongasome dynamics experiments.

BILLAUDEAU, C., CHASTANET, A., YAO, Z., CORNILLEAU, C., MIROUZE, N., FROMION, V. & CARBALLIDO-LOPEZ, R. 2017. Contrasting mechanisms of growth in two model rod-shaped bacteria. *Nat Commun*, 8, 15370.

DION, M. F., KAPOOR, M., SUN, Y., WILSON, S., RYAN, J., VIGOUROUX, A., VAN TEEFFELEN, S., OLDENBOURG, R. & GARNER, E. C. 2019. *Bacillus subtilis* cell diameter is determined by the opposing actions of two distinct cell wall synthetic systems. *Nat Microbiol*.

SUN, Y., HÜRLIMANN, S. & GARNER, E. 2023. Growth rate is modulated by monitoring cell wall precursors in *Bacillus subtilis*. *Nature Microbiology*.

ZIELIŃSKA, A., SAVIETTO, A., DE SOUSA BORGES, A., MARTINEZ, D., BERBON, M., ROELOFSEN, J. R., HARTMAN, A. M., DE BOER, R., VAN DER KLEI, I. J., HIRSCH, A. K. H., HABENSTEIN, B., BRAMKAMP, M. & SCHEFFERS, D.-J. 2020. Flotillin-mediated membrane fluidity controls peptidoglycan synthesis and MreB movement. *eLife*, 9, e57179.

Reviewer #3 (Remarks to the Author):

There is much to like in this new study of the dynamics of the *B. subtilis* elongasome. The authors considerably extend previous studies done by TIRF microscopy by orienting cells vertically to allow detection of movement over longer distances and times than was possible previously. This single-molecule “VerCINI” approach reveals long processive movements of MreB that frequently reverse direction, pause, and restart. Notably, the reversals and pauses were much more frequent than observed before, likely because of limitations of the TIRFm approach. Additional measurements by stroboscopic illumination reveal a relatively long lifetime of MreB subunits within filaments, and allow measurements of photobleaching. The smVerCINI results indicate relatively long lifetimes of processive and paused motility states. The processivity at constant velocity is now determined to correspond approximately to half the diameter of the cell. Interestingly, these movement parameters are not strongly changed in cells grown under different conditions. A second series of experiments shows that modulating RodA glycosyltransferase amounts changes the bidirectional motility and processivity, with speed and processivity increasing at lower RodA amounts. These results are interpreted in terms of “tug-of-war” models in which multiple elongasome complexes bind to MreB filaments. Another implication of this work is that RodA amount is limiting in *B. subtilis* cells growing under these conditions, whereas the cognate Class B PBPs seem to be in excess. In the last part of this paper, the authors model different versions of the “tug-of-war” model. This modeling favors an end-binding model over a multiple binding model. Remarkably, the end-binding model can account for the observation that cell diameters increase at low and high levels of RodA induction, which has not been accounted for before.

This is a succinct, innovative paper that provides new insights into the elongasome dynamics of the experimental system used. The paper is convincing and the data are convincing as far as it goes. The paper will be of considerable interest and impact to the field. Significantly, the paper critically revives the “tug-of-war” model, which seems to fit with this new data. There are a couple points about the cell physiology that need to be mentioned and clarified in the paper. In addition, to help readers not in this area, the authors might include some definitions of some parameters and brief heuristic explanations of some of the methods. Some suggestions and comments follow.

1. The paper strongly shows the importance of cellular concentrations for elongasome components, in this case RodA, on outcomes of these dynamics experiments. The motion determinations are of a previously reported MreB-HT fusion reporter. More should be said about this MreB-HT fusion here. What was its cellular concentration compared to the WT MreB protein?

We have carried out a Western blot to show expression levels of MreB-HaloTag, which we find is comparable to wild-type levels (Supplementary Figure 1a, referenced line 97). We also see negligible cleavage between MreB and the HaloTag.

Was it fully functional in the absence of the other MreB homologs? If the MreB-HT amount or activity is not WT, how would this affect the interpretation of the experiments?

We have performed additional characterization of the MreB-HaloTag fusion. In particular, we confirmed that the MreB-HaloTag fusion can support growth in PAB medium, where we previously

found that MreB is essential for survival [7] (Supplementary Figure 1b). We compared the growth rate of wild-type strain PY79 to the MreB-HaloTag fusion strain and show it has a minimal effect on cell growth rate (Supplementary Figure 4a). We show that morphology of PY79 and SM01 (*mreB-halotag, Δhag::erm*) is similar (Figure 1c). We also show that addition of excess magnesium – which is known to compensate for MreB/Mbl/MreBH defects [8] - does not affect cell morphology (Figure 1c). Together with the Western blot quantification mentioned above, these data demonstrate that the MreB-HaloTag fusion is highly functional.

SI Figure 1 is now referenced in line 97.

2. Along a similar line, what is known about the relative stoichiometry of RodA compared to PBP2A and PBPH in *B. subtilis* cells? It seems that this must be known and should be discussed as context.

This is an important point to clarify. It has not yet been possible to determine precise stoichiometry of RodA to PBP2A and/ or PBPH due to current lack of suitable antibodies for quantitative Western blotting.

In previous work by the Garner lab, overproduction of PBP2A/H transpeptidases alone do not affect cell width, whereas RodA overproduction increases cell width, again suggesting transpeptidases to be already in excess and not a limiting factor on elongasome dynamics [6]. Also, our data focusing on single deletions of each of the two transpeptidases (PBP2A, PBPH) shows little effect on MreB dynamics. If either of the transpeptidases were limiting, we would see a detrimental effect on elongasome dynamics.

We have expanded our discussion of this point in the results section (Lines 247-256):

“We also tested how single knockouts of the redundant elongasome transpeptidases PBP2A (*pbpA*) and PBPH (*pbpH*) affected elongasome dynamics. Deletions of these genes had little effect on elongasome switching kinetics, speed or processivity (SI Fig. 4). This is consistent with previous observations where overexpression of individual elongasome transpeptidases alone did not affect cell width [2]. In addition, the elongasome transpeptidase in *E. coli* is produced to levels where most of the protein is freely diffusive and not bound to the elongasome [19]. Together, these results support a model where the elongasome transpeptidases are in excess compared to RodA, and thus not the limiting factor in elongasome activity. Our data show that RodA concentration, or its assembly to an elongasome complex, is the principal factor controlling the concentration of active elongasome synthesis complexes within the cell any given time.”

To further test the idea that RodA is limiting, does overexpression of PBP2A and/or PBPH affect MreB velocity?

This is an interesting point but even if overexpression of PBP2A/H did affect elongasome dynamics it would not change the core conclusion that tug-of-war regulates elongasome activity.

In follow up work which is outside of the scope of this study we plan to delve further into molecular mechanism of tug-of-war including which components are rate limiting, how individual elongasome components turn over from the main complex etc, which will address the sorts of points raised by the reviewer about elongasome stoichiometry.

Last, were the extent of RodA under- and overexpression (e.g., Fig. 2f) determined by western blotting? Does the level of RodA amount in the controlled strain correspond to the WT level when the WT speed and processivity are observed.

This is a valid point. Currently, there are no antibodies for RodA to carry out a western blot to quantify RodA levels, and RodA is quite a tricky protein to purify and work with.

In response to this reviewer comment, we made extensive efforts to attempt to quantify RodA protein levels using mass spectrometry. However, unfortunately this yielded inconclusive results – which is not uncommon for low concentration membrane proteins - and we were unable to quantify RodA levels in any strain including wild-type. Therefore, we are at this time unfortunately unable to quantify changes in RodA concentration upon *rodA* over/underexpression. However, given that we see the expected major changes in cell shape which are characteristic of RodA over/underexpression it is likely that the changes in RodA concentration are substantial.

We hope to address this in future either by adding a hopefully functional C-terminal FLAG tag to RodA or by raising antibodies against the protein.

3. Line 106 and Fig 1d-e. A bit more might be explained about the stroboscopic illumination experiments.

We have now further explained the stroboscopic illumination experiment in the results section. Lines 114-157 have been revised as:

‘We determined the binding lifetime of MreB subunits within filaments and the JF549 photobleaching lifetime using the stroboscopic illumination method of Gebhardt and coworkers [14]. We measured the apparent binding lifetime of MreB molecules by single molecule tracking and systematically increased the total time interval between frames (strobe interval), while keeping the exposure time and thus total light dose constant (Fig. 1d-e). As the strobe interval increased, the fraction of time for which the molecule was illuminated decreased thus increasing the effective photobleaching lifetime. This not only allows molecules to be tracked for longer but also allows accurate determination of both molecule binding lifetime time and dye photobleaching lifetime by fitting an equation describing the relationship between those quantities, the strobe interval, and the apparent binding lifetime (Fig. 1e, Methods).

We measured the MreB binding lifetime as 128 s [95% CI: 109, 164] showing that the MreB filaments remain assembled at the membrane for extended periods of time. This measurement represents a lower bound on the lifetime of both assembled MreB filaments, as it likely represents a combination of MreB filament unbinding from the membrane, dissociation of MreB subunits from the MreB filament and occasional migration of the elongasome complexes beyond the microscope depth of field. We also measured the JF549 bleaching lifetime as 13s [95% CI: 11, 16].

We next characterized MreB motility by smVerCINI. We chose a strobe interval of 6 s, which extended the effective photobleaching lifetime 12-fold to 156 s [95% CI: 132, 192] (Fig. 1e), longer than the median observed MreB subunit lifetime of 128s [95% CI: 109, 164]. This allowed direct measurements of MreB single molecule switching kinetics. We found that MreB filaments are motile 81 % of the time, [Range: 79-81, $n=3$], and immobile (paused) the

rest of the time (Fig. 1i). MreB filaments frequently switch between motile and paused states, and motile molecules frequently change direction, where the leading edge of the filament becomes the lagging edge, and peptidoglycan synthesis commences in the opposite direction (reversal) (Fig. 1i-k), SI Fig 10d). The median lifetimes of both the processive and paused motility states were substantial: 40.5 s [95% CI: 39.0, 43.0] and 27.0 s [95% CI: 24.0, 29.5], respectively (Fig. 1f). ‘

4. Line 137. Add “(Fig. 1f)” at end of the line.

We added this.

5. Fig. 1c is very hard to read because of its size and color, and Fig. 1i is not drawn properly (see Supplemental Fig. 3).

We increased the size of Figure 1c. We cannot identify the issue with Figure 1i, but we are happy to address this later if the reviewer can clarify.

6. Some definitions or brief explanations for some terms would help improve clarity, including processive subtracks, circumferential track displacement, elongasome displacement, track density, processive and static track lifetimes.

This is a helpful suggestion. We have added Supplementary Table 1, which defines the terms we use to describe elongasome dynamics and how they are measured.

7. Fig. 3. In the end-binding model, are both synthases actively synthesizing in opposite directions or is only one synthase actively synthesizing and the other binding the other end, as depicted? It would seem that both synthases would be active in opposite directions to get a tug-of-war effect. Please clarify.

Based on the MKL simulated model, during processive motion only one synthase is active. The oppositely bound synthase stochastically attempts to initiate synthesis in the opposite direction – biologically this could correspond for example to the bPBP of the complex binding to the existing cell wall. The entire elongasome then enters a brief tug-of-war situation where both synthases are stalled. This is resolved either by continuation of synthesis in the original direction or reversal – ie PG synthesis termination and initiation of synthesis in the opposite direction. The predicted brief tug-of-war associated pause is too fast to detect currently.

We have further described this in Supplementary Figure 10.

8. Along the same line, can synthase complexes (e.g., RodA-sfGFP) be detected at the ends of MreB-HT(red) filaments by SIM-TIRF in two-color experiments? This seems like a relatively simple experiment that would provide direct and compelling evidence for the end-binding model, and would indicate if unlimited binding does occur at all.

This is an excellent experimental proposal, which we are planning to carry out in future studies investigating the detailed molecular mechanisms of tug-of-war. However, it is not a relatively simple experiment. Rather, it is a very tricky one which is right at the limits of resolution of the SIM technique – MreB filaments are about 100 nm long, equal to SIM resolution, and one would need to

resolve the position of individual elongasome synthesis complexes along those filaments. Therefore, it is out of scope of the current study.

We think that our current results already demonstrate our core finding that tug-of-war is regulating elongasome synthesis activity and bidirectional dynamics, whereas the detailed molecular mechanisms underlying this process await further study.

9. Fig. 4b. The conclusion is no difference in track density, but the graph seems to indicate a difference. What do the statistics (P values) say? Likewise, in Fig. 4a and in other comparisons, the trends and distributions look very nice based on large numbers of determinations, but are the differences statistically different. Please clarify this point here and elsewhere.

We should clarify that in Figure 4B we were talking about there not being any difference in MreB filament density between 10uM and 1000uM IPTG induction of *rodA* (where there is no detectable difference) rather than 0.1 uM IPTG vs the other conditions, which we think the reviewer is talking about. However, we notice that for Figure 4B we did not originally make statistical comparisons – we have now carried this out (Supplementary Table 6, Lines 351-355).

However, we need to push back a bit on the reviewer's request for P values. We certainly do perform quite careful and extensive statistical comparisons between various experimental conditions, mentioned throughout the text and summarised in SI Table 5. However, we consistently use the well established Estimation Statistics approach (see eg Introduction to the New Statistics, Cummings & Calin-Jageman 2017) rather than null hypothesis significance testing (NHST, aka stating the p-values).

The key concept here is rather than state a p-value for a comparison between two conditions, one should state the effect size, ie the observed difference between the two conditions, and the estimated uncertainty on the observed difference. Eg (Lines 223-224): "MreB pausing rate decreased 0.43-fold (-0.13 min⁻¹ difference [95% CI: -0.17,-0.10])"

This has the advantage over NHST of analysing not just the probable reproducibility of a difference between two conditions, but also the estimated magnitude of the difference between the two conditions. This is useful because it allows assessment of whether magnitude of difference is large enough to be physically/ biologically meaningful, rather than just reproducible. There are many cases in biology where a perturbation causes a significant, reproducible that is so small that it is irrelevant. Estimation statistics has been developed to address this issue. It has already seen wide uptake in medicine, and is rapidly being adopted in biology. See eg https://en.wikipedia.org/wiki/Estimation_statistics for further discussion of its history and uptake. We note that multiple learned societies have already called for NHST to be abandoned and estimation statistics are one of the most common and widespread alternatives.

For comparison to NHST analysis, we note that for a median/ mean observed difference where the 95% CIs are both greater or less than zero this would indicate rejection of the null hypothesis that the two population medians/ means are equal at p = 0.05 significance level.

In Supplementary Table 6, we have added statistical analysis comparing processive MreB filament density with RodA depletion and overexpression.

We have clarified the specific conditions we are comparing. (Lines 351-355):

‘We found that cell widening upon *rodA* overexpression (1 mM IPTG) is not associated with any detectable change in surface density of motile MreB filaments compared to induction with 10 μ M IPTG (0.36 track density μm^{-2} difference [95% CI: -0.54, 1.3]) (Fig. 4b, SI Table 6), which have previously been shown to regulate cell width [6], nor any detectable change in cell growth rate (SI Fig. 12).’.

We have added the following rationale for using this method rather than P values in the methods section (Lines 632-643):

‘Effect size estimates [9] were calculated based on difference of medians, using either DABEST (Data Analysis with Bootstrap Coupled Estimation [10]) or custom bootstrapping scripts. All effect sizes are listed in Supplementary Table 7. In comparison to null hypothesis significance testing (NHST) analysis, which only reports on whether a difference may exist between two populations, effect size analysis addresses one of the flaws of NHST by allowing analysis of not only whether a difference may exist but also the magnitude of that difference[9], which is often of critical importance for interpreting the biological significance of a result. For comparison to NHST analysis, a median/ mean observed difference whose 95% CIs are both greater or less than zero this would indicate rejection of the null hypothesis that the two population medians/ means are equal at $p = 0.05$ significance level.’

10. Line 331. Please consider qualifying a bit. More needs to be done to definitively support the “tug-of-war” model (see point 8, above), but the case made here is strong for support of this model.

We respectfully disagree with the reviewer on this point. While the detailed mechanisms of tug-of-war remain to be determined (see below), all our data fit the tug of war model, and the various possible alternative models (discussed Lines 238-246) are incompatible with the data. We have now also shown that lipid II levels are unlikely to affect elongasome dynamics in our experiments (see Reviewer 2 rebuttal, major comment #2).

However we fully agree that the detailed molecular mechanism of elongasome tug-of-war remains to be determined. Is tug-of-war mediated by end-binding vs unlimited binding or some other intermediate model, and how does turnover of elongasome components regulate tug-of-war? This is what we will investigate over the next few years with planned – difficult! - experiments such as those raised in point 8 above.

We have rephrased the statement in the discussion to (Lines 388-392):

“We showed that elongasome processivity and bidirectional motility are regulated by molecular motor tug-of-war between multiple elongasome synthesis complexes. Our data also provide initial evidence for an end-binding tug-of-war between maximally two synthesis complexes bound to opposite ends of the antiparallel MreB filament (Figure 5a).”

We also make the following statement at the end of the discussion (Lines 403-405):

“Further studies will be required to determine the detailed molecular principles underlying elongasome tug-of-war and to conclusively determine the role and extent of molecular motor tug-of-war in regulating bacterial cell shape.”

11. Supplemental Fig. 1. How do the growth parameters compare to the WT strain that lacks MreB-HT? There is no comparison here (please see point 1, above).

Please see response to Reviewer 3 comment 1.

Rebuttal References

- [1] Y. Sun, S. Hürlimann, E. Garner, Growth rate is modulated by monitoring cell wall precursors in *Bacillus subtilis*, *Nat. Microbiol.* 8 (2023) 469–480.
<https://doi.org/10.1038/s41564-023-01329-7>.
- [2] J. Domínguez-Escobar, A. Chastanet, A.H. Crevenna, V. Fromion, R. Wedlich-Söldner, R. Carballido-López, Processive Movement of MreB-Associated Cell Wall Biosynthetic Complexes in Bacteria, *Science* 333 (2011) 225–228.
<https://doi.org/10.1126/science.1203466>.
- [3] J. Domínguez-Escobar, A. Chastanet, A.H. Crevenna, V. Fromion, R. Wedlich-Söldner, R. Carballido-López, Processive Movement of MreB-Associated Cell Wall Biosynthetic Complexes in Bacteria, *Science* 333 (2011) 225–228.
<https://doi.org/10.1126/science.1203466>.
- [4] E.C. Garner, R. Bernard, W. Wang, X. Zhuang, D.Z. Rudner, T. Mitchison, Coupled, Circumferential Motions of the Cell Wall Synthesis Machinery and MreB Filaments in *B. subtilis*, *Science* 333 (2011) 222–225. <https://doi.org/10.1126/science.1203285>.
- [5] A. Vigouroux, B. Cordier, A. Aristov, L. Alvarez, G. Özbaykal, T. Chaze, E.R. Oldewurtel, M. Matondo, F. Cava, D. Bikard, S. van Teeffelen, Class-A penicillin binding proteins do not contribute to cell shape but repair cell-wall defects, *eLife* 9 (2020) e51998.
<https://doi.org/10.7554/eLife.51998>.
- [6] M.F. Dion, M. Kapoor, Y. Sun, S. Wilson, J. Ryan, A. Vigouroux, S. van Teeffelen, R. Oldenbourg, E.C. Garner, *Bacillus subtilis* cell diameter is determined by the opposing actions of two distinct cell wall synthetic systems, *Nat. Microbiol.* 4 (2019) 1294–1305.
<https://doi.org/10.1038/s41564-019-0439-0>.
- [7] H. Strahl, F. Bürmann, L.W. Hamoen, The actin homologue MreB organizes the bacterial cell membrane, *Nat. Commun.* 5 (2014). <https://doi.org/10.1038/ncomms4442>.
- [8] B. Tesson, A. Dajkovic, R. Keary, C. Marlière, C.C. Dupont-Gillain, R. Carballido-López, Magnesium rescues the morphology of *Bacillus subtilis* mreB mutants through its inhibitory effect on peptidoglycan hydrolases, *Sci. Rep.* 12 (2022) 1137.
<https://doi.org/10.1038/s41598-021-04294-5>.
- [9] G. Cumming, R. Calin-Jageman, *Introduction to the new statistics: Estimation, open science, and beyond*, Routledge, 2016.
https://books.google.com/books?hl=en&lr=&id=KR8xDQAAQBAJ&oi=fnd&pg=PP1&ots=1mrxLbuFZ2&sig=ubArAc2_GAOKQNfgt0fbXPM1z6A (accessed February 23, 2024).
- [10] J. Ho, T. Tumkaya, S. Aryal, H. Choi, A. Claridge-Chang, Moving beyond P values: data analysis with estimation graphics, *Nat. Methods* 16 (2019) 565–566.
<https://doi.org/10.1038/s41592-019-0470-3>.

Reviewer #1 (Remarks to the Author):

As already stated in my previous report, this is a beautiful experiment together with an original modeling-supported interpretation. In my opinion, the authors have very nicely answered all questions raised by the reviewers and clarified the open points. I am strongly in favor of accepting the paper in its revised form.

Reviewer #2 (Remarks to the Author):

I have read the revised version of this paper with great interest and would like to thank the authors for their careful consideration of the comments made during the first round of review. I was initially very enthusiastic about this paper, which in my view has now been improved by the clarifications provided and the additional controls performed to exclude that substrate availability plays a role in the observed dynamics. I have no further comments and recommend that this paper is accepted.

Reviewer #3 (Remarks to the Author):

The authors did an outstanding job addressing each point raised in the previous reviews, which were uniformly positive. In many cases, the authors included more explanation of approaches and broadened their definitions of terms. They also included new data to meet comments from the previous reviews. Last, they tightened up and refined certain interpretations. Their responses to all three reviews are well reasoned and clear. They made considerable effort to modify the previous version, and these changes resulted in improvements to an already compelling story.

The overall outcome of these revisions is an important, well-written paper with several novel and interesting results about the mechanism of PG elongation in rod-shaped bacteria. This paper is a major contribution to this field because of its findings, explanations, conclusions, and advances in methodology.

There are only two cosmetic issues that the authors can attend to quickly.

1. Often, compound adjectives like single-molecule and multidrug-resistant lack hyphens. Please correct throughout.
2. The pastel or light-colored text in many of the figures is hard to read, even on the computer (e.g., Fig. 1j and 1k; Fig. 2d; especially Fig S1c labels; Fig. S6g and S6h; Fig. S7f and S7g). Please use darker colors.

Response to reviewers' comments

REVIEWERS' COMMENTS

Reviewer #1 (Remarks to the Author):

As already stated in my previous report, this is a beautiful experiment together with an original modeling-supported interpretation. In my opinion, the authors have very nicely answered all questions raised by the reviewers and clarified the open points. I am strongly in favor of accepting the paper in its revised form.

Reviewer #2 (Remarks to the Author):

I have read the revised version of this paper with great interest and would like to thank the authors for their careful consideration of the comments made during the first round of review. I was initially very enthusiastic about this paper, which in my view has now been improved by the clarifications provided and the additional controls performed to exclude that substrate availability plays a role in the observed dynamics. I have no further comments and recommend that this paper is accepted.

Reviewer #3 (Remarks to the Author):

The authors did an outstanding job addressing each point raised in the previous reviews, which were uniformly positive. In many cases, the authors included more explanation of approaches and broadened their definitions of terms. They also included new data to meet comments from the previous reviews. Last, they tightened up and refined certain interpretations. Their responses to all three reviews are well reasoned and clear. They made considerable effort to modify the previous version, and these changes resulted in improvements to an already compelling story.

The overall outcome of these revisions is an important, well-written paper with several novel and interesting results about the mechanism of PG elongation in rod-shaped bacteria. This paper is a major contribution to this field because of its findings, explanations, conclusions, and advances in methodology.

There are only two cosmetic issues that the authors can attend to quickly.

1. Often, compound adjectives like single-molecule and multidrug-resistant lack hyphens. Please correct throughout.

We have corrected this throughout.

2. The pastel or light-colored text in many of the figures is hard to read, even on the computer (e.g., Fig. 1j and 1k; Fig. 2d; especially Fig S1c labels; Fig. S6g and S6h; Fig. S7f and S7g). Please use darker colors.

We have used darker colours to make the figures easier to read.